# Fuzzy Rules to Help Predict Rains and Temperatures in a Brazilian Capital State Based on Data Collected from Satellites

**Paulo Vitor de Campos Souza** [1,*,†] , **Lucas Batista de Oliveira** [2,†] ,
**Luiz Antônio Ferreira do Nascimento Jr.** [3]

1   Federal Center of Technological Education of Minas Gerais-CEFET-MG-Information Governance Secretariat, Belo Horizonte 30421-169, Brazil
2   Faculty Una of Betim-Information Systems Course, Betim 32510-010, Brazil; lobatista@outlook.com.br
3   Graduate Program in Electrical Engineering-Federal University of Minas Gerais-UFMG, Belo Horizonte 31270-901, Brazil; lafnjr@gmail.com
*   Correspondence: pauloc@prof.una.br
†   These authors contributed equally to this work.

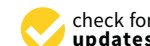

**Featured Application: A hybrid model formed by neural network techniques and fuzzy systems that predicts rainfall and temperatures in the capital of Minas Gerais, Brazil, which contains several river sources essential for the Brazilian economy. The purpose of the paper is to construct a model capable of extracting knowledge from satellite data related to temperatures and rainfall forecast of the analyzed state.**

**Abstract:** The forecast for rainfall and temperatures in   underdevelope countries can help in the definition of public and private investment strategies in preventive and corrective nature. Water  is an essential element for the economy and living things. This study had a main objective to use an intelligent hybrid model capable of extracting fuzzy rules from a historical series of temperatures and rainfall indices of the state of Minas Gerais in Brazil, more specifically in the capital. Because this is state has several rivers fundamental to the Brazilian economy, this study intended to find knowledge in the data of the problem to help public managers and private investors to act dynamically in the prediction of future temperatures and how they can interfere in the decisions related to the population of the state. The results confirm that the intelligent hybrid model can act with efficiency in the generation of predictions about the temperatures and average rainfall indices, being an efficient tool to predict the water situation in the future of this critical state for Brazil.

**Keywords:** fuzzy neural network; rain forecast; temperature prediction

## 1. Introduction

Meteorology has always been an element of great importance for humanity due to the various activities that depend on the climatic condition. A good example is agriculture [1], as the climate is one of the main factors determining the productivity of a crop; it is also used for predicting natural disasters such as floods and droughts [2]. These events cannot be inhibited, but when anticipated, it can contribute to a premeditated plan of action, avoiding further damage.

The rainfall frequency, unfortunately, has been suffering interference, from the evolutionary actions of the human race [3]. In recent years, it is common to find reports of large cities declaring a state of emergency and rationing for lack of water, as occurred recently in the capital of Belo Horizonte in the state of Minas Gerais, which carries the title of water tank in Brazil, because it is the cradle of

watersheds critical nationally. Along with several great cities, it has suffered a strong impact from drought [4].

In contrast to the lack of water, it has problems with excess: floods and landslides [2]. If a river or stream receives a considerable amount of water from rain and cannot support it, it ends up overflowing and causing floods, which bring with it destruction, damage, and even death.

The facts cited above are just some of the impacts that rain has on society. Within this context, this work was developed for the use of neural networks in the forecast of volumetric rainfall, based on a fuzzy rule system, developing a precise forecast, and contributing to the anticipation of possible tragedies s in places that occur phenomena of greater magnitude, as well as helping in a better control of agriculture. Another purpose was to support air and sea transportation, avoiding accidents in their trajectories. Briefly, when anticipating a possible problem, it is also possible to anticipate solutions and reduce their impacts, such as evacuating areas at risk of landslides or flooding [5].

Currently, many kinds of research are being developed to apply models of artificial neural networks in rain forecasts, river basins, and water studies. For example, it was applied to the problem of forecasting the flow of the Nile River in Egypt [6], using a temporal series as a reference to compare tests among several prediction methods of neural networks. Besides, there was a study using stochastic autoregressive mean models in time series for rainfall forecasting in the Apennines mountains [7], Italy. Another exciting study was carried out due to the need for greater precision in the prediction of time series. In this model, the researchers used a hybrid neural network on monthly flow data from the Colorado River at Lees Ferry, USA [8]. ANN was also applied to predict the short-term flow of the Winnipeg River in Northwest Ontario, Canada. The developed ANNs consistently outperformed conventional models during the test phase for all four predictions [9]. Another study presented a new procedure to identify the structure and parameters of three-layer ANN models and demonstrate the potential of such models to simulate hydrological behavior, providing a better representation of the rain in river basin flow near Collins, Mississippi, USA [10]. Lastly, the importance of rainfall prediction in water studies used a genetic algorithm coupled to ANN to simulate a rainfall field, implemented to predict rainfall several times using rain gauge in the Upper Watershed Parramatta, in the western suburbs of Sydney, Australia [11]. Thus, it is not an innovation in water studies and meteorology, which has always been side-by-side with the technology. However, when dealing with neural networks, there is a range of possibilities and paths. This paper presents a path that it contributes to the studies already developed. The use of intelligent models to aid in the prediction of meteorological aspects is vital to assist government managers and people connected to the industry. However, in many cases, the models used are not self-explanatory, thus allowing the results obtained to become of complex. Therefore, intelligent models that are capable of bringing interpretable results are fundamental for the dissemination of knowledge about a particular scientific context. The fuzzy neural networks act as a model capable of bringing the interpretability to the results extracted from a database. These models should act in synergy to obtain positive results that are easy to understand [12].

This work proposes the use of the fuzzy neural network proposed by de Campos Souza and Bambirra [13] to act in the Brazilian meteorological system dataset in the forecast of rains and temperatures in the capital city of Minas Gerais, which has a high number of river sources and can generate shortages in the country if the volumes of rainfall or the temperature increases uncontrollably. It uses the concept of fuzzy rules based on equally spaced membership functions and fuzzy neurons for the construction of a fuzzy inference system. Connected to this system is a neural aggregation network that can perform the necessary approximation of the expected responses.

The main novelty of this paper is the extraction of fuzzy rules based on real data of temperature and rainfall in the Belo Horizonte, in addition to the use of a hybrid model to act efficiently in predicting the best results. This is the first work in the literature that seeks to predict rainfall and temperature indices in this crucial state of Brazil. This paper proposes the efficiency of the model in fuzzy rules extraction. Prediction tests were performed with intelligent models that are commonly used in the literature for predicting rains and temperatures.

The remainder of the paper is organized as follows: In Section 2, the main concepts and themes presented in the paper are presented to the reader, including aspects related to the state of Minas Gerais and concepts about meteorology and intelligent models. In Section 4, we discuss aspects of the architecture and the model used in the paper for extracting knowledge from the database. In Section 5, the tests are described, the methodology used, and the models chosen to compare the results. In Section 6, the conclusions and the expected future works are presented to the reader.

## 2. Literature Review

### 2.1. Minas Gerais State

Minas Gerais is a governmental unit of Brazil, being the fourth largest state with the second most inhabitants and voters, located in the Southeastern Region of the country [14]. It is a state that has universal relevance for having been one of the regions that most provided gold for Europe. Minas Gerais is bordered by states that are relevant to the Brazilian economy, such as Sao Paulo, Rio de Janeiro, and Bahia. As the second-largest contributor to Brazilian GDP, this southeast state has in its territory several economic and cultural characteristics [15], such as the historic cities of Mariana and Ouro Preto, listed as world heritage and at the same time the country's largest road network.

Most of the territory of Minas Gerais has altitudes that oscillate between 900 and 1500 m with a predominance of plateaus with cliffs and depressions [16].

Concerning its hydrographic bases, the state of Minas Gerais has among its principal rivers: *The Doce River*, which rises between the slopes of the Mantiqueira and Espinhaço ranges [17] and flows into the Atlantic Ocean [18], in the Brazilian state of Espirito Santo; *The Grande River*, whose source is in the Serra da Mantiqueira [19], in the municipality of Bocaina de Minas until reaching the Paranaíba River, thus forming the Paraná River in the state of São Paulo; *The Paranaíba River*, which is born in the Wood of the Cord, in Paranaíba, and has approximately 1070 km of extension; and *São Francisco River*, which rises in the Serra da Canastra, cutting through the state of Bahia and passing through northeastern states like Pernambuco, Sergipe and Alagoas until it empties into the ocean, and itswater are essential for tourism, leisure, irrigation, and transportation in several cities, especially in northern Minas Gerais [20].

It is also worth mentioning the Jequitinhonha River, which is born in the Serra do Espinhaço, in Serro, and travels a great distance until its mouth in the Atlantic Ocean [21]. Figure 1 presents a satellite view of the Minas Gerais territory, highlighting the main rivers and hydroelectric plants present in the territory. Figure 2 presents a satellite image showing the accumulation of water vapor over the state of Minas Gerais in July 2019.

Belo Horizonte is the capital of the state of Minas Gerais, being the municipality of Minas Gerais responsible for the highest economic indices, and concentrations of people, services and trade. It currently owns companies that control climate-related aspects and weather stations throughout the state and is the city with connections to abundant water sources and four water containment systems http://www.copasa.com.br/wps/portal/internet/abastecimento-de-agua/nivel-dos-reservatorios responsible for most of the water supply of the Minas Gerais state [22].

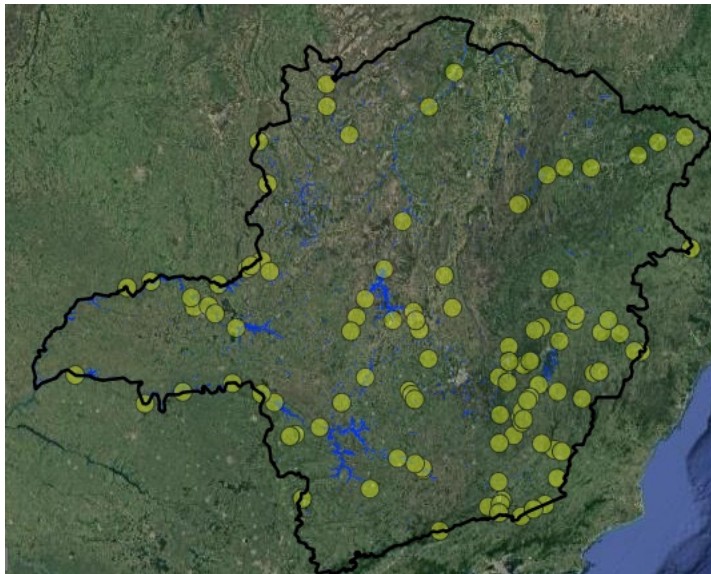

**Figure 1.** The territory of Minas Gerais. Water concentration and hydroelectric plants (yellow dots). Source: http://idesisema.meioambiente.mg.gov.br/.

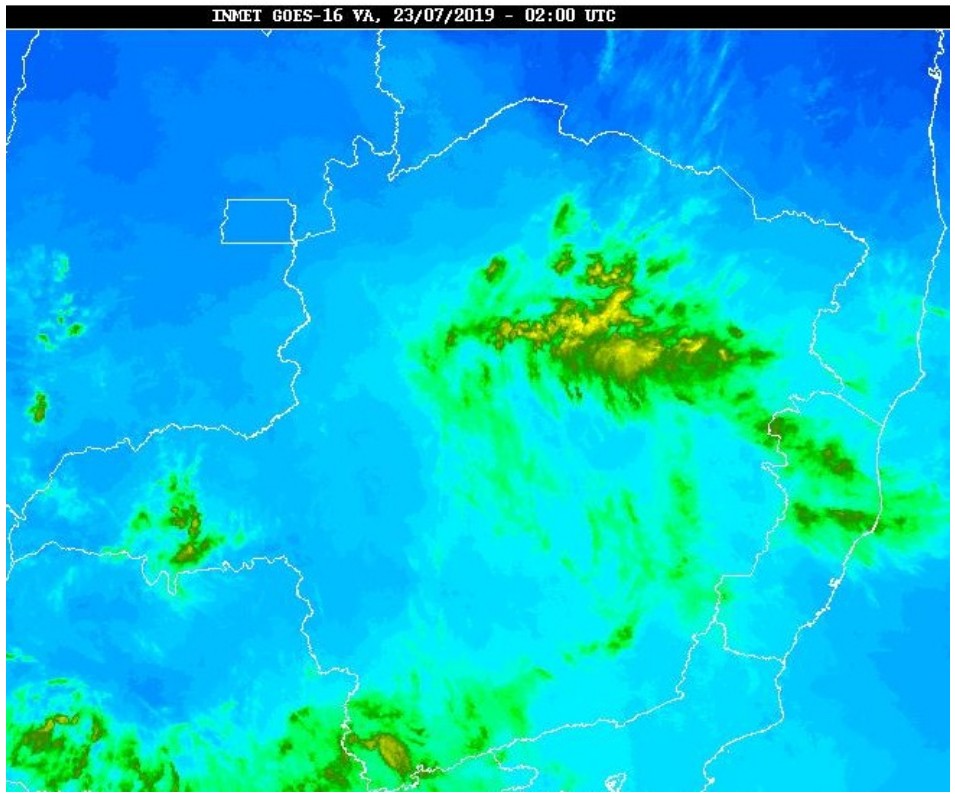

**Figure 2.** Water vapor over Minas Gerais in July 2019 (satellite image). Adapted from Inmet.

## 2.2. Meteorology

Meteorological control in the state of Minas Gerais, as well as throughout Brazil, allows assessing the impacts present in the environment [23]. These assessments are supported by the collection of pertinent data from components present in the environment that may directly affect social, economic, and environmental aspects of the territory of the state of Minas Gerais.

Therefore, knowing climate change, natural phenomena, and impacts resulting from these events can foster public policies, business investments, and cutting-edge research to harness natural

phenomena and produce sustainability in the environment or new resources to improve the local economy [24]. Figure 3 exhibits characteristics of the behavior of wind currents (200 hPa, in July 2019) present in the Brazilian territory and, consequently, their impacts on the analyzed state.

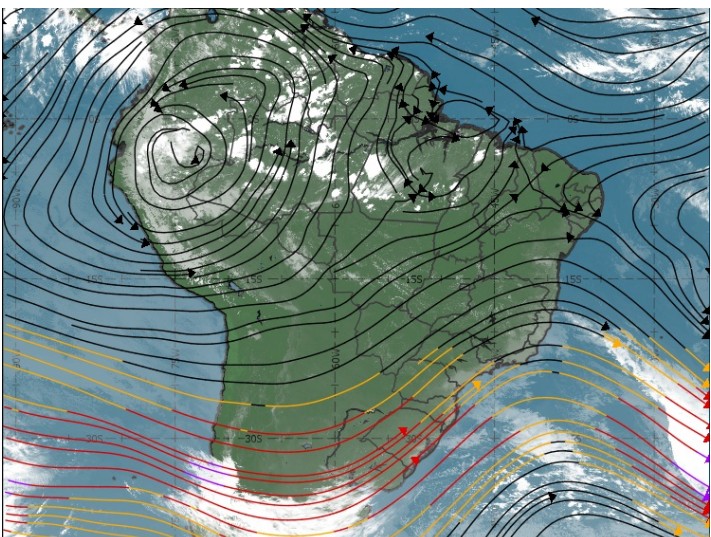

**Figure 3.** Wind currents behavior (200 h Pa) (satellite Image). Adapted from Inmet.

*2.3. Weather Forecast*

A weather forecast (also called prognosis) is the application of science and technology to make a detailed description of expected future occurrences in the atmosphere at a specific location. A weather forecast includes the use of models based on certain atmospheric parameters [25]. In addition, it involves the search based on the momentary meteorological conditions that seek to anticipate future weather conditions. This study involves knowledge and data from a variety of areas, such as sky views, cloud formation analysis, observed temperature, atmospheric pressure, Doppler radar images, meteorological stations, atmospheric balloons, and other data [26].

From the knowledge and data collected from several areas [27], monitoring of air masses occurs 24 h a day. Information about temperature, air pressure, wind speed, and humidity are also collected [28]. These data are used to make numerical models that run on supercomputers, which compare the initial state and analyze the various possibilities of the evolution of time, generating predictions based on probabilities [29]. Even with all this reading done by modern computers, much of the forecast comes from the readings made by meteorologists, who analyze the predictions and conclude on them after making the corrections that are necessary [30].

The climatic behavior of Minas Gerais is a relevant factor for the culture, economy, and agricultural activities in this territory. The state concentrates a considerable number of river springs vital to Brazil, a large part of the territory destined to the breeding of animals, coffee plantations, and other food types. Therefore, it is necessary to evaluate the climate behaviors in the state, allowing corrective or preventive measures to be taken to avoid associated impacts on daily actions in Minas Gerais. These meteorological indices are monitored by agencies controlled by the Brazilian and Minas Gerais government. There are physical stations to control state-critical regions, receiving data, and climate specifications from each of these regions. Figure 4 presents installations of these climate control structures in southeastern Brazil, mainly focusing on the territory of the state of Minas Gerais.

For example, rainfall and weather influences can change actions and strategies in food planting or in storing water for people or crops [31]. Figure 5 presents an image collected from a satellite, identifying in July 2019 (traditionally with low rainfall) locations and densities of water precipitation in the Brazilian territory.

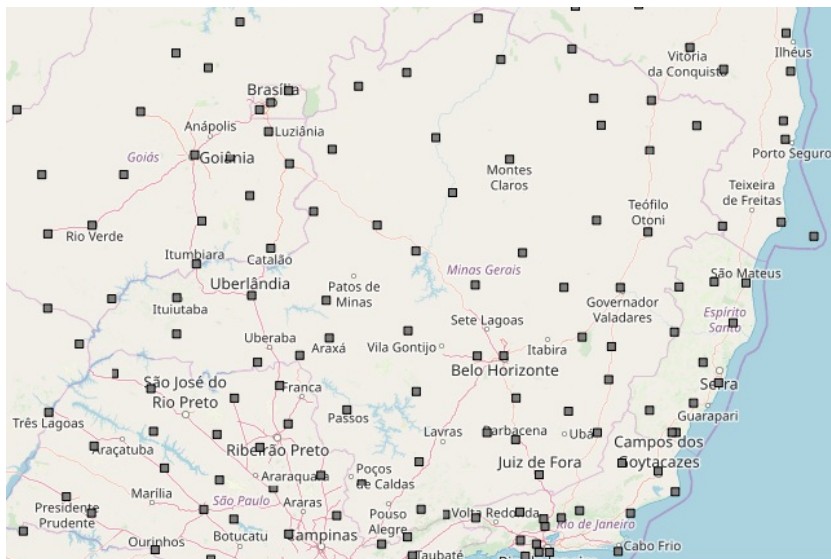

**Figure 4.** Meteorological stations installed in southeastern Brazil. Adapted from Inmet.

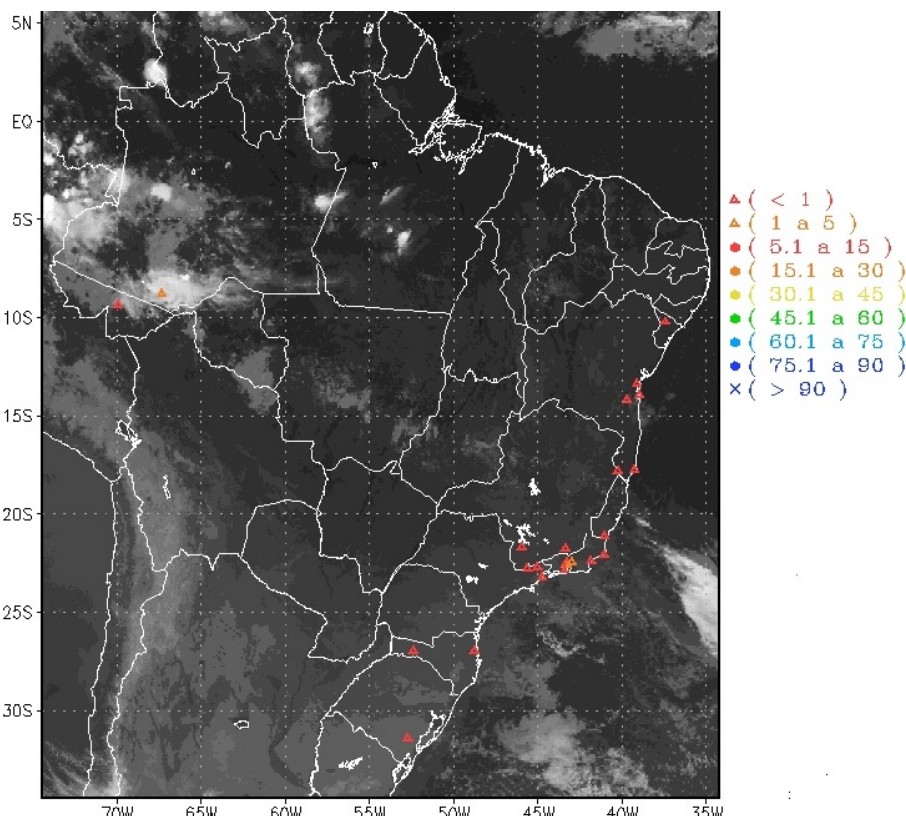

**Figure 5.** Rainfall indices (mm/h) in the month of July 2019. Adapted from Inmet.

Another relevant factor for climate analysis can be observed in assessing a nation's temperatures. Figure 6 represents an assessment of average temperatures in Brazil in August 2019.

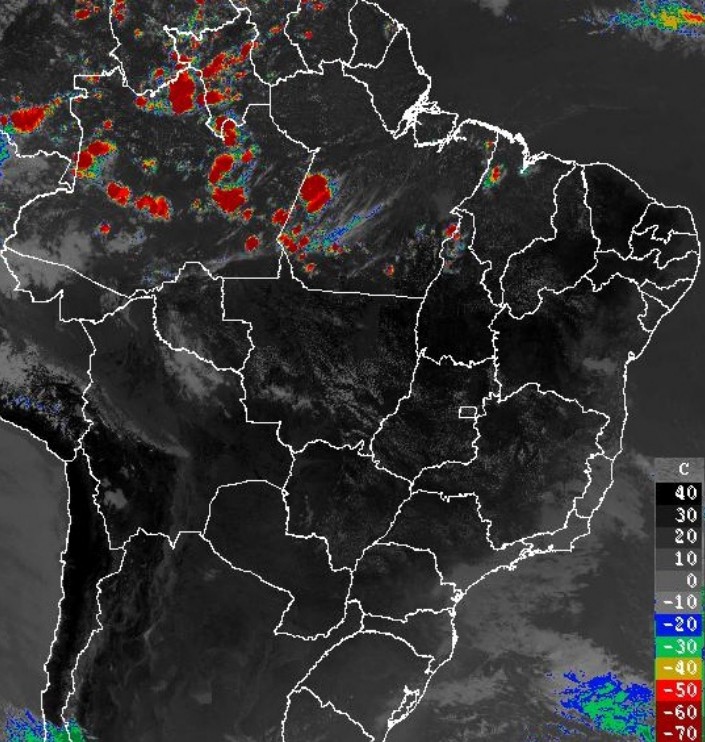

**Figure 6.** Brazilian temperatures in August 2019. Adapted from Inmet.

*2.4. Extraction and Collection of Meteorological Data*

Meteorological data are collected to assist decision-making by administrative management to assess human, sustainability, economic, and environmental aspects linked to rainfall and climate change elements. Devices, sensors, and satellites to capture environmental reactions and allow the organization of data of several natures. Climate control agencies are responsible for collecting and storing data of various natures, such as satellite images, precipitation data, reservoir level, etc. In Brazil, the main entity responsible for controlling the meteorological aspects is the National Institute of Meteorology (Inmet) http://www.inmet.gov.br and in the state of Minas Gerais stands out the control of water reservoirs for the Minas Gerais Basic Sanitation Company (COPASA) http://www.copasa.com.br/wps/portal/internet. Historical data from a state can help in understanding climate behavior to estimate historical series on rainfall, temperatures, and relative humidity. The represent the behavior of climatic aspects are presented in Figures 7–10 elements about average temperatures, humidity, evaporation, and precipitation, respectively, between 1931 and 1960, comparing them with the periods from 1961 to 1990, month by month. It is noticeable that some behaviors follow a trend in this assessment, but some changes due to external factors (such as global warming) altered historical behavior during the months evaluated.

When evaluations of the average temperature (Figure 7) of the State of Minas Gerais was introduced, it is possible to identify peak temperatures in January and February. Similarly, the lowest average temperatures were obtained in June and July. Another factor to highlight is that the average temperature in the second evaluation period shows their gradual increase in the state of Minas Gerais.

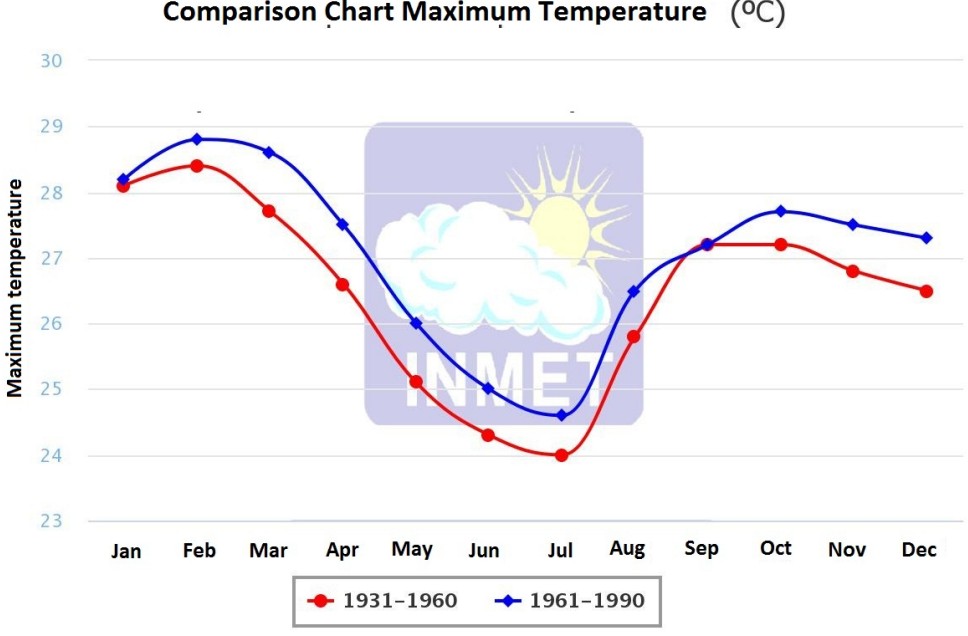

**Figure 7.** Minas Gerais temperatures: historical series. Adapted from Inmet.

In the graph (Figure 8) that expresses the humidity of this Brazilian state, there is no similar pattern between the two periods present in the figure. Through the months evaluated, the first phase of evaluation had higher humidity than the second period of analysis in three months (March, April, and December). Another factor to be highlighted is that the least humid months in this state of southeastern Brazil are the months of July, August, and September. On the other hand, the highest humidity is observed in the first month of the year.

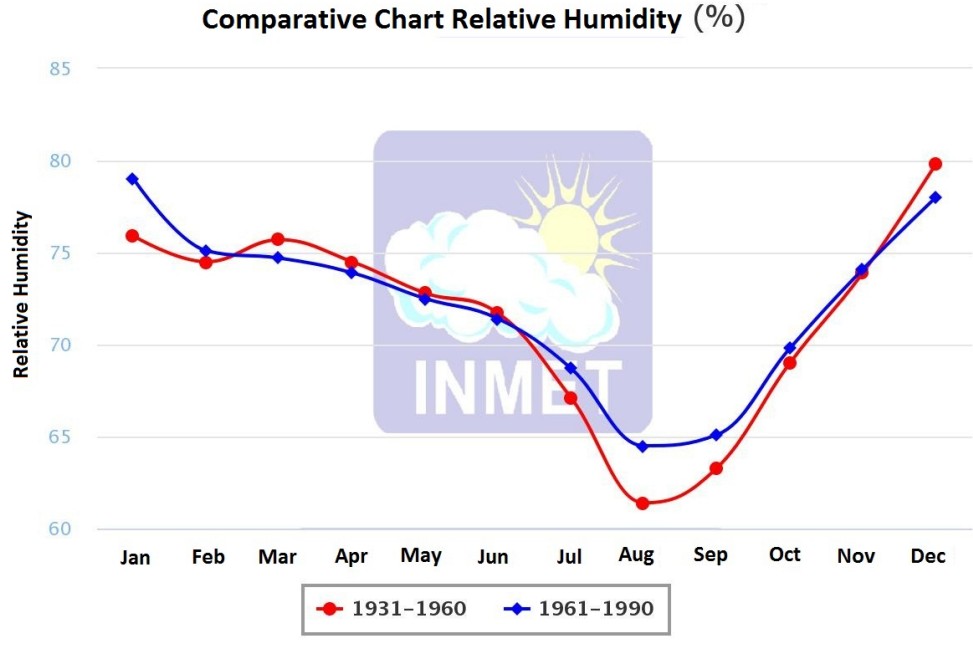

**Figure 8.** Minas Gerais humidity: historical series. Adapted from Inmet.

Figure 9 expresses the evaporation behavior present in Minas Gerais. In August and September, the highest evaporation rates were found in both periods. In addition, it is observed that, from

October, the indices decrease rapidly. In the second period analyzed, there is more excellent stability of evaporation rates, especially from March to June. In the analysis that begins in the 1930s, the measures obtained in the graphical analysis are lower for six months and equal for another two months when compared to the period ending in the 1990s.

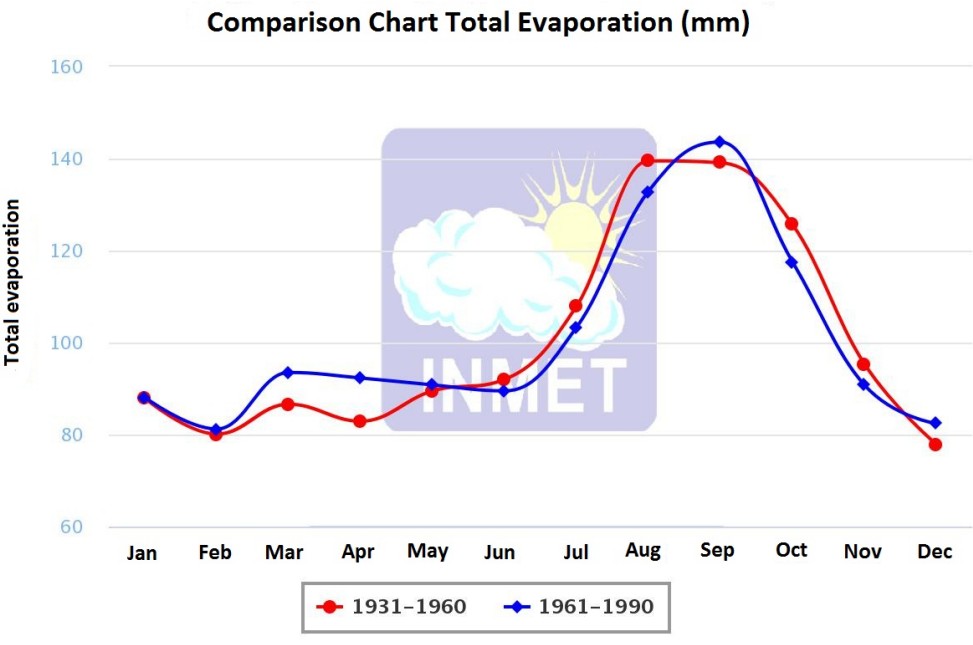

**Figure 9.** Minas Gerais evaporation: historical series. Adapted from Inmet.

Finally, Figure 10 shows the average rainfall of Minas Gerais. Both periods exhibit remarkably similar behavior. The months of December and January stand out as the wettest periods. On the other hand, the central months of the year (from May to September) represent the periods with scarce rainfall.

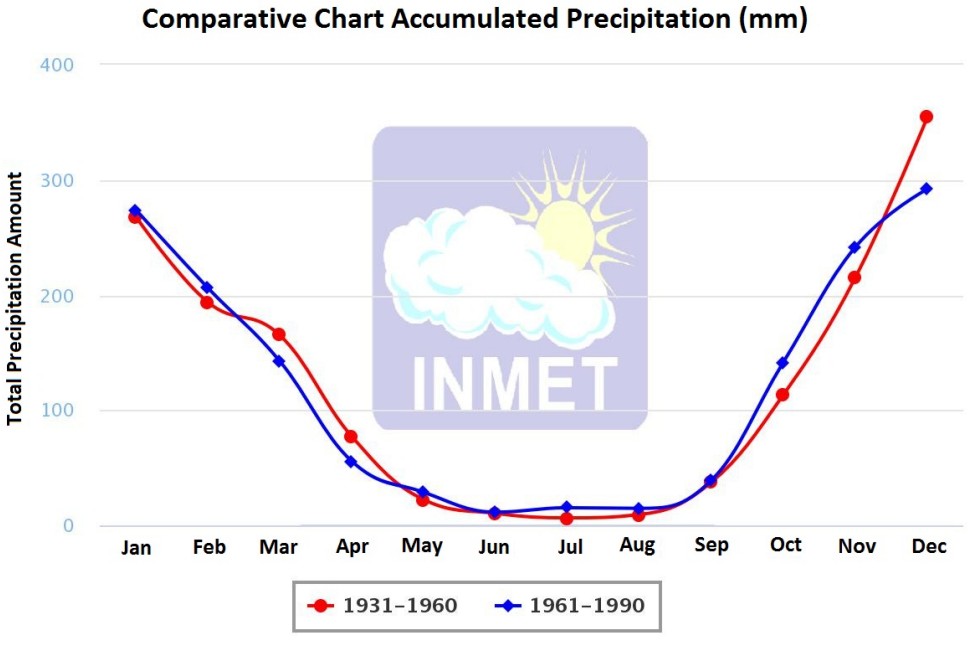

**Figure 10.** Minas Gerais precipitation: historical series. Adapted from Inmet.

The collection of historical data allows us to identify behavioral changes and situations that may generate disturbances and specific adjustments in economic activities. The reduction of rainfall and rising temperatures can have a decisive role in decision making by managers or rulers. Figures 11 and 12 exhibit a change in values linked to the environment, especially precipitation and average temperatures. In this state, there are warnings issued over large mines that can rupture and generate environmental problems, as reported in Brumadinho in 2019 [32] and November 2015 in the historic city of Mariana [33]. These disasters cost the lives of numerous people and severely contaminated springs and rivers, critical to the water supply of the state's citizens. Therefore, determining techniques to predict climate change is fundamental for this state relevant to Brazil for its rivers and water sources that are fundamental for consumption, transportation, and economic activities.

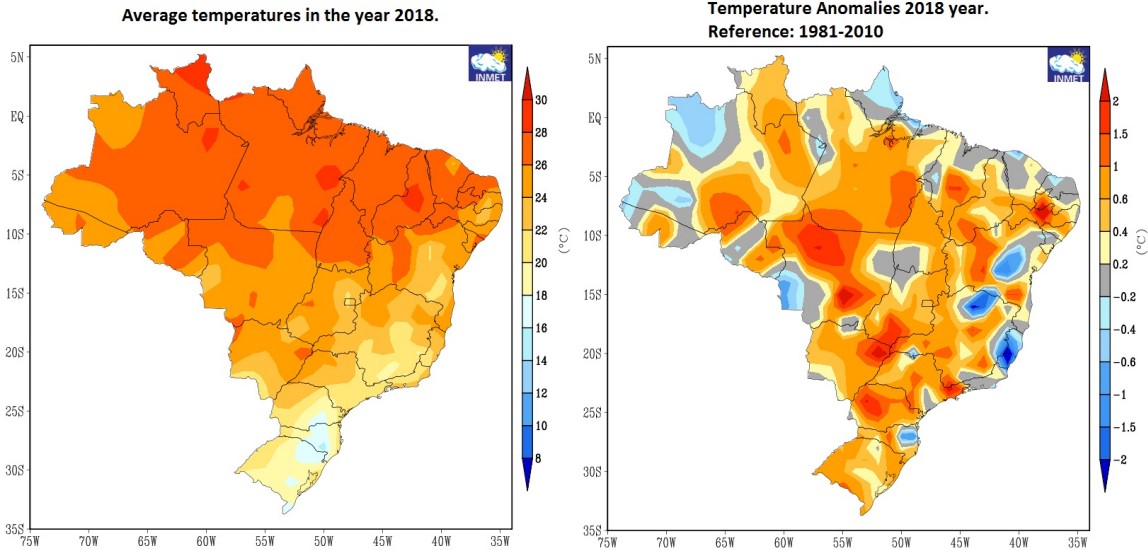

**Figure 11.** Comparison between climate change and anomalies in 2018 in Brazil. Adapted from Inmet.

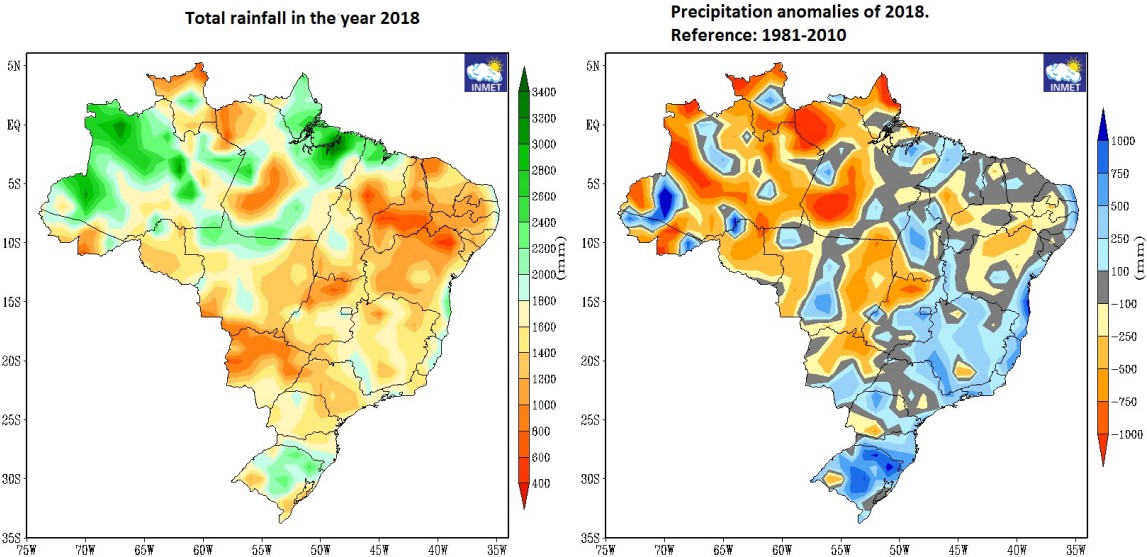

**Figure 12.** Comparison between rainfall and anomalies index in 2018 in Brazil. Adapted from Inmet.

Different factors are hugely affected by the dry climate and lack of rainfall associated with burns brought out in the state's vegetation. Brazil has been suffering from several fire outbreaks, mainly due to winter and lack of rainfall. Figure 13 shows possibility of fire occurrence through Nesterov

flammability index [34] in Brazil in August 2019. It is possible to see in the map the number of points in the state of Minas Gerais with a high probability of fires in vegetation environments.

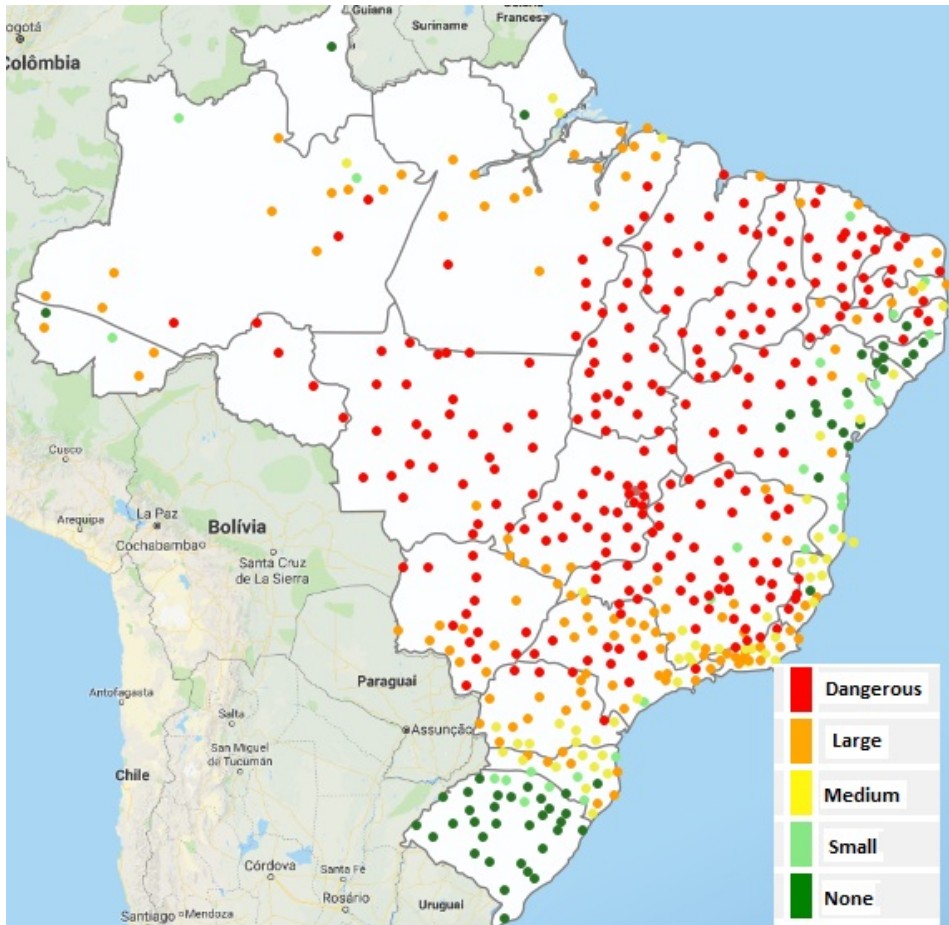

**Figure 13.** Probabilities of burning in the Brazilian territory in August 2019. Nesterov flammability index. Adapted from Inmet.

## 2.5. Artificial Neural Network and Fuzzy Systems

Artificial neural networks (ANN) are one of the main tools used in machine learning. As the "neural" part of the name suggests, the networks are brain-inspired systems, which are intended to replicate the way that humans learn. Neural networks consist of input and output layers as well as (in most cases) a hidden layer consisting of units that transform the input into something that the output layer can use. Most of the time, the ANNs can act as nonlinear models [35].

ANNs are computational systems modeled after the biological neural networks present in the animal kingdom. Neural networks learn to perform tasks without being told explicitly to do certain things. They find a wide range of applications from language processing to image processing. Artificial neural networks are excellent tools for finding patterns that are far too complex or numerous for a human programmer to extract or to teach a machine to recognize. While neural networks have been around since the 1940s, it is only in the last several decades that they have become a significant part of artificial intelligence [36].

Fuzzy logic [37] is a form of multi-valued logic in which the truth values of variables may be any real number between 0 and 1, i.e., $[0, 1] \in$. In other words, it uses the concept of partial truth for uncertainty [34], where the truth value may range between completely true and completely false. In contrast, Boolean logic captures the truth values of variables, which may only be the integer values 0 or 1. Since Zadeh (1965) [38] introduced the concept of the fuzzy sets; and several mathematical concepts such as number, group, topology, and differential equation have been fuzzified. Several

extensions and types of fuzzy set theory have been proposed to solve the problem of constructing membership degree functions of fuzzy sets and to represent the uncertainty associated with the considered problem in a way different from the fuzzy set theory [39]. One of the greatest advantages of fuzzy systems lies in the interpretability as they can perform inference with human-readable knowledge expressed in the form of fuzzy if-then rules [40]. The clear presentation of knowledge helps users to gain insights into the complex problems and to facilitate the explanation of their solutions.

Prof. Zadeh proposed that, to model this vagueness of the linguistic information, similar to human cognition, the semantics of the linguistic information can be represented using the fuzzy sets (FSS) [38]. Prof. Zadeh proposed the concept of FSs in 1965 through his seminal work. FSs model the uncertainty of the linguistic information through membership functions (MFs).

### 2.6. Intelligent Models Acting in the Prediction of Temperatures and Rains

The use of advanced science to assist in the prediction of rainfall and temperature has developed for many years. The model of Hsu et al. [10] acts on the use of artificial neural networks in the construction of intelligent models for rain prediction. On the other hand, the model by Poff et al. [41] has been using smart models since 1996 to help predict climate change worldwide. Another intelligent type assists in features of economics through the prediction obtained on climate change, as proposed by Sailor et al. [42].

Similarly, more recent studies highlighting the same themes are the works of Ghose and Samantaray [43], Esteves et al. [44], and Graham et al. [45]. Dhar et al. took the deep learning approach [46] to aid in the quantitative prediction of rainfall. Intelligent hybrid models were additionally used for the same purposes reported above by Lee and Liu [47] to weather forecasting, and an adaptive model for rain forecasting in Malaysia was created by El-Shafie et al. [48]. It is also worth noting the recent model proposed by Ashrafi [49] for forecasting rainfall and river routing indices and the hybrid structure suggested by Soares et al. [50] that works with the weather forecast series prediction in four Brazilian capitals (Sao Paulo, Manaus, Porto Alegre, and Natal). Therefore, there are no correlated works in the forecast of rainfall and temperatures in cities of the state of Minas Gerais, especially of its capital, using hybrid models.

## 3. Materials and Methods

The dataset used in this research are available on the INMET website http://www.inmet.gov.br/portal/index.php?r=bdmep/bdmep. Therefore, all techniques and methodologies for data collection and collection are adequately explained on the website of the Brazilian institution.

### Feature of the Database

The primary analyses were performed with emphasis on the monthly interpretation of the meteorological data of the state of Minas Gerais. The filters for choosing the data are available as follows:
-State: MG.
-Attributes: Predominant Wind Direction, Average Wind Speed, Maximum Average Wind Speed, Tar Evaporation, BH Potential Evapotranspiration, BH Real Evapotranspiration, Total Sunstroke, Average Cloudiness, Number of Days with Precipitation, Total Precipitation, Mean Sea Level Atm Pressure (mbar), Mean Atm Pressure (mbar), Mean Maximum Temp (°C), Mean Compensated Temp (°C), Mean Maximum Temp (°C), Mean Relative Humidity, and Average Visibility.

In order to be able toaccess the data and perform the appropriate consultations, it is necessary to make a registration with personal information.

The data resulting from the attributes selection are correlated to all cities in the state of Minas Gerais that have data collection stations (49 stations).

To determine meteorological factors in the Minas Gerais, we highlight the station that collects data from the most relevant region of the state, the capital Belo Horizonte. This is because the city addresses

a region of 7–10 municipalities relevant to the Minas Gerais economy, in addition to having three of the main dams that store drinking water for the consumption of a significant portion of citizens of the state (Electronic address for monitoring the reservoir levels of water in the metropolitan region of Belo Horizonte: http://www.copasa.com.br/wps/portal/internet/abastecimento-de-agua/nivel-dos-reservatorios). Table 1 presents the relevant data from the station where the data were collected.

**Table 1.** Data station information.

| Station: Belo Horizonte—MG (OMM: 83587) | Value |
|---|---|
| **Latitude (degrees):** | $-19.93$ |
| **Longitude (degrees):** | $-43.93$ |
| **Elevation (meters):** | 915.00 |
| **Start of operation:** | 03/03/1910 |
| **Requested period of data:** | 01/01/2000 to 04/01/2019 |

The initial results selection process sought to perform an analysis of missing information in the data provided. Therefore, all dimensions that had incomplete data were discarded in the construction of the database. All dimensions that have data collected for the months analyzed were the target of this research. Table 2 summarizes the data resulting from this evaluation process; where applicable, the standard deviation is given in parentheses.

**Table 2.** Data Station Features and values.

| Station: Belo Horizonte—MG | Range or Average |
|---|---|
| **Mount** | 1 to 12 |
| **Year** | 2000 to 2019 |
| **Wind Direction** | 9.128 (5.324) |
| **Average Wind Speed** | 1.458 (0.314) |
| **Total Insolation** | 202.438 (40.407) |
| **Average Cloudiness** | 4.862 (1.705) |
| **Mean Pressure** | 913.511 (2.251) |
| **Mean Maximum Temperature** | 27.463 (1.819) |
| **Mean Compensated Temperature** | 22.166 (1.784) |
| **Mean Minimum Temperature** | 18.138 (1.933) |
| **Mean Relative Humidity** | 62.429 (7.49) |

Figures 14 and 15 show the organization of the database after selecting records that did not meet the initial criteria. In summary, there are complete records on the climate aspects of the city of Belo Horizonte from January 2000 to April 2019.

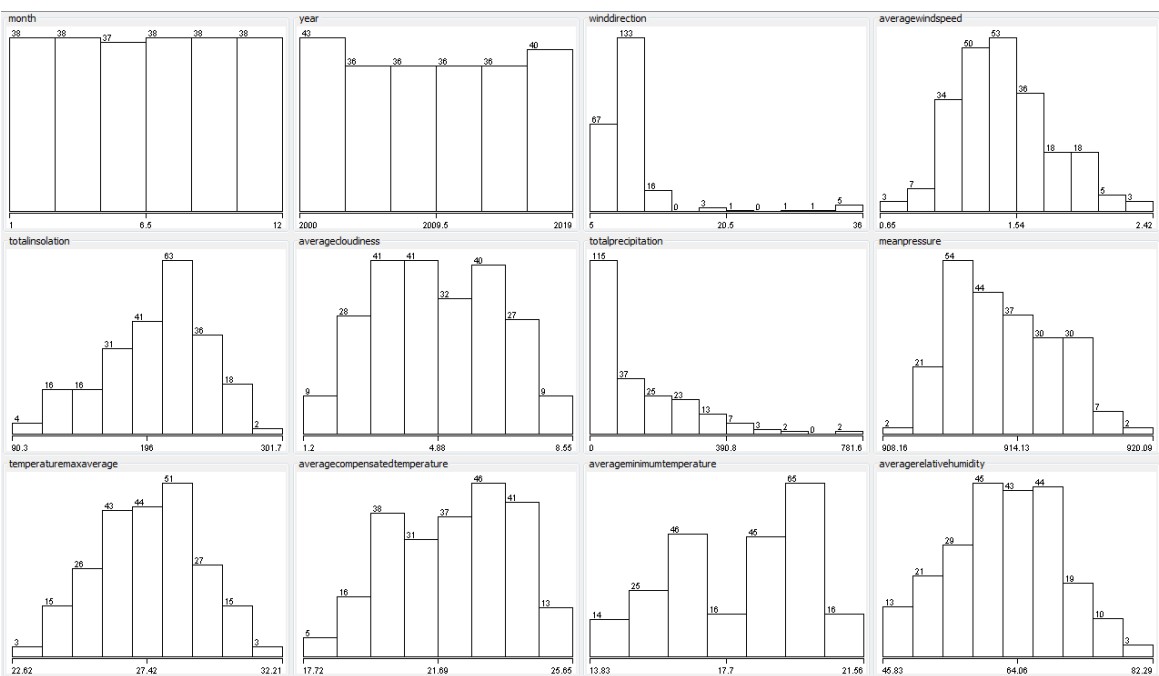

**Figure 14.** Dataset: General data and its main characteristics.

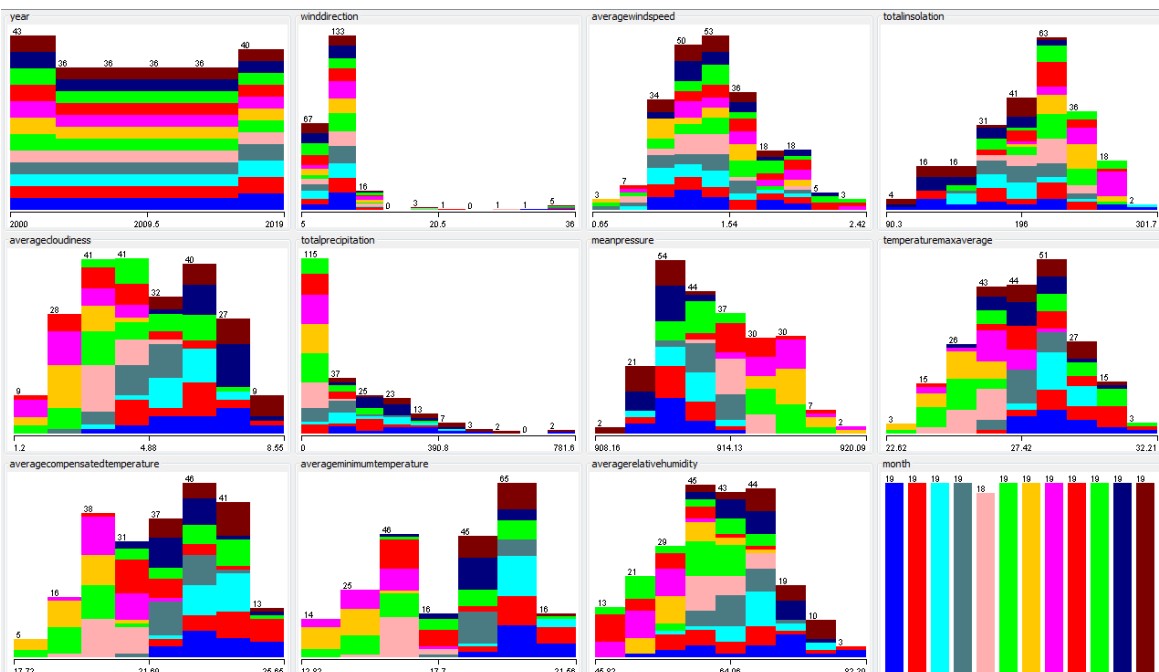

**Figure 15.** Dataset: General related data by month.

## 4. Fuzzy Neural Network for Predicting Rains and Temperatures

This paper presents the application of a fuzzy neural network to behave in climatic perspectives in the state of Minas Gerais. To evaluate aspects of climate, data collected at the environmental station in the state capital, Belo Horizonte, were evaluated. This site was chosen four of the largest water reservoirs are in the metropolitan region of the state. This region contributes a large part of the state's GDP and has many people, businesses, and companies needing a large amount of water. Finally, another factor to be highlighted is the proximity of these dams to the fluvial regions affected by disasters of mining ditches (Mariana and Brumadinho). Figure 16 presents the three layers present in the hybrid model used in this study.

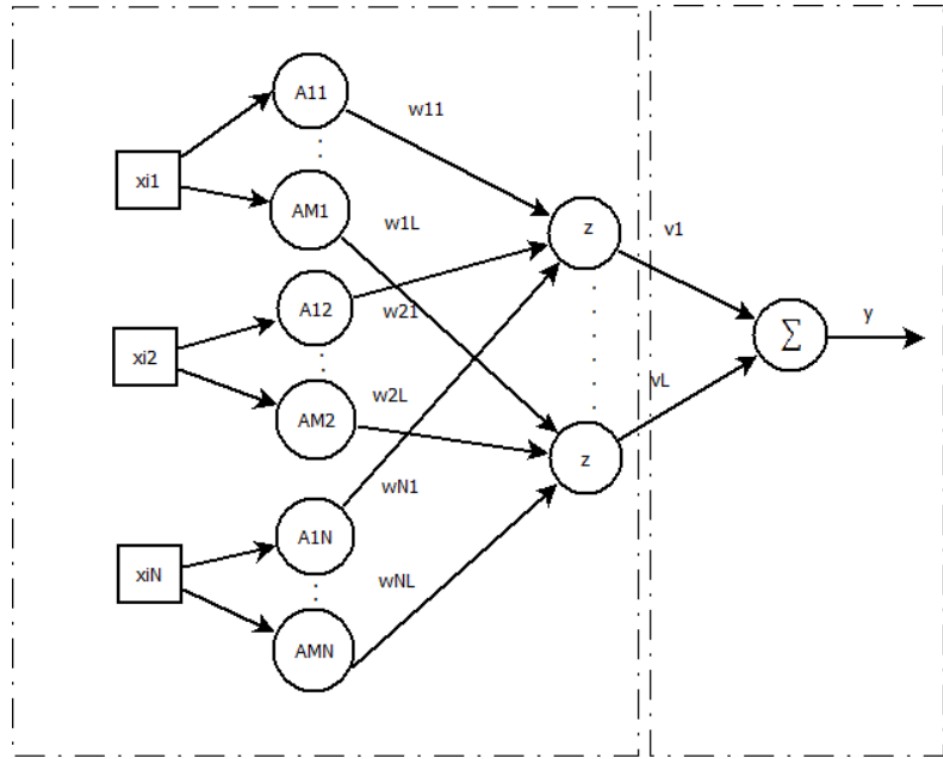

**Figure 16.** Fuzzy Neural network [13].

### 4.1. Fuzzy Neural Networks

Fuzzy neural networks are hybrid models designed using two intelligent techniques generally applied in problem-solving, highlighting the training methodologies of neural networks and the ability to extract knowledge and interpretability of problems arising from fuzzy systems [51].

Fuzzy neural network models produce architectures similar to the Multilayer Perceptron [52] networks, but with the experience to acquire knowledge from the database submitted to the model. FNNs are structured in layers, where each has specific employment. Usually, the first layers are responsible for the fuzzification process, converting inputs into fuzzy sets. Inner layers are capable of performing and/or fuzzy aggregating rules. Finally, the last layers of the model can transform fuzzy numbers into traditional numerical values [53]. The main highlight of these hybrid models, making them different from other artificial intelligence models, is their expertise in building fuzzy rules that can lead to expert problem-solving systems [54].

These hybrid models have an extensive area of expertise in several economic and social areas. Since the 1990s, heterogeneous models have been employed for frame control [55], pattern recognition applications [56], and in power electronics and motion control [57]. Already in the 2000s, a FNN was used to solve problems such as delta wing control [58], motion control systems [59], and financial reporting fraud risks [60]. In addition, in health problem solving, Lim et al. [61] used a fuzzy neural network for real-time premature ventricular contraction detection and Wang et al. [62] employed the ability of intelligent hybrid models to detect features of cardiac arrhythmia. Finally, in the last decade, scientific work using such techniques has addressed detection of celestial materials (called Pulsars) [63]; prediction of autism in children [64], adolescents [65], and adults [66] through database obtained from mobile devices; breast cancer [67]; absenteeism at work [68]; control for active power filter [69,70]; data knowledge [71]; and speech recognition [72]. The diversity of these models also covers the field of predicting software building efforts [73], cybersecurity [74], help in cryotherapy and immunotherapy treatments [75,76], and different classification and regression problems [77–83], where the models differ according to the training techniques, fuzzification or defuzzification process, architecture, number of layers, elements present in the model structure, etc.

### 4.2. First Layer

In the initial layer of the model, procedures presented by Jang [84] are applied to generate uniformly *M* spaced membership functions of the triangular type (calculated through center 0.5). Thus, the neurons of the first layer represent the fuzzification method of the problem data under analysis. Therefore, these neurons, whose activation functions are membership functions of fuzzy sets, granulate the input space to form a fuzzy partition (see, e.g., the work of De Campos, Souza, and Bambirra [63]).

The Adaptive Network-Based Fuzzy Inference System (ANFIS) [84] consists of five succeeding layers capable of establishing the membership functions of the problem decision space. The first layer of ANFIS is responsible for determining fuzzy membership functions through adaptive parameters. This layer can be expressed by [84]:

$$O_i^1 = \mu_{A_i}(x), \qquad i = 1, 2 \ldots, n \tag{1}$$

where $x$ is input value, $\mu$ is the membership degree, and $O_i^1$ is membership value of fuzzy variable $A_i$ [84]. Another relevant factor for understanding the model is in the second layer of the ANFIS, where every node in this layer is a fixed node which acts as a product operation, as in Sugeno fuzzy model [84]:

$$O_i^2 = W_i = \mu_{A_i}(x) \times \mu_{B_i}(y), \qquad i = 1, 2 \ldots, n \tag{2}$$

where $B_i$ is a fuzzy variable. These varieties of groups, through membership functions with the equivalent features, can assist in accurate analysis of the problem. This procedure performs a non-linear mapping of its data space to the output area. This mapping is accompanied by several fuzzy IF/THEN rules, where individually one relates the bounded execution of the mapping. The antecedents in the rules of a fuzzy inference system perform a multidimensional neural division, which can be a grid, decision tree, or cluster. In this paper, the method works with a limitation of 200 neurons in the first layer to circumvent the problem of high dimensionality. This decision is adequate for numerous preceding experiments that verified that the ability to create fuzzy rules made the issue much more complicated than was essential. For each input variable $x_{ij}$, $L$ neurons are defined $A_{lj}$, $l = 1, \ldots L$, whose activation functions are composed of membership functions in the corresponding neurons. Thus, the expected result in the first layer of the model is the degrees of association related to the inputs submitted to the model, [63]:

$$a_{jl} = \mu_{A_l}, \, j = 1 \ldots n, \, l = 1 \ldots L \tag{3}$$

where, for each ANFIS input result, the number of problem inputs and the number of fuzzy sets are, respectively, $n$ and $L$ [63]. This layer stands out in the paper because it is responsible for extracting primary knowledge from the database evaluated by the model. Thus, it is possible to identify suitable relations between dimensions of the problem and membership functions.

The following equation defines the triangular functions ($\triangle$) used in the construction of the first layer triangular fuzzy neurons:

$$\triangle(x, \nu, \vartheta, \zeta) = max\left(min\left(\frac{x-\nu}{\vartheta-\nu}, 1, \frac{\zeta-x}{\zeta-\vartheta}, 0\right)\right) \tag{4}$$

which is defined by three parameters for defining three points: $\nu$ and $\zeta$ for the base, and $\vartheta$ for the extremity of the triangular fuzzy neuron.

### 4.3. Second Layer

Unineurons [85] constitute the second layer of the FNN. They apply the concepts of a fuzzy operator uninorm [86] to implement simplified operations according to the activation function of the fuzzy neurons. Its construction allows the unineuron to practice both concepts of: "neuron *and*" and

"neuron *or*". It can be seen as a mapping that stretches triangular norms by providing the identity element to be a value in the unit interval. Because of this, it has the properties of commutativity, associativity, and monotonicity, as well as its identity element. The processing of neurons occurs at two levels. At the first level of $L_1$ sections, the input signals are connected individually in addition to the weights. In the second, at a global level of $L_2$, a global aggregation operation is executed on the results of all first-level sequences. These structures are excellent operators able to perform universal approximation of functions through fuzzy system modeling [87]. This layer performs the aggregation of the $L_c$ neurons from the first layer within the unineurons introduced by Lemos et al. [85]. In this paper, it is expressed as follows:

$$U(x,y) = \begin{cases} o.\,T(\frac{x}{o}, \frac{y}{o}), & if\ y \in [0,\ o] \\ o + (1-o)\,.S\,(\frac{x-o}{1-o}, \frac{y-o}{1-o}), & if\ y \in (o,\ 1] \\ \beta(x,y), & otherwise \end{cases} \tag{5}$$

$$\beta(x,y) = \begin{cases} max\ (x,y) & ,if\ o \in [0,\ 0.5] \\ min\ (x,y) & ,if\ o \in (0.5,\ 1] \end{cases}' \tag{6}$$

where $T$ is a *t-norm* (algebraic product), $S$ is a *s-norm* (probabilistic sum), and $o$ is the identity element. When $o = 0$, the uninorm is the orneuron type [88], and, when $o = 1$, the uninorm is the andneuron type [88].

The unineuron proposed by Lemos et al. [85] performs operations to unify existing fuzzy values. This process can be seen as a preliminary process for turning two values into one and is defined by the following two steps:

(1) Each pair $(a_i, w_i) = b_i = \mathbf{p}\ (a_i, w_i)$.
(2) Unified aggregation $=\mathbf{U}\ (b_1, b_2 \ldots b_n)$, where $n$ is the number of inputs.

The function $p$ (relevancy transformation) is responsible for transforming the inputs and respective weights into individual transformed values. This function accomplishes the four condition: *monotonicity in value*, *zero importance elements should have no effect*, *normality of importance of one*, and *consistency of effect of the weight*. In order for a $p$ function to meet all four necessary requirements of the relevancy transformation operator, Yager proposed the formulation [89]:

$$p(w,a,o) = w.a + \bar{w}.o \tag{7}$$

where $\bar{w}$ represents the complement of $w$. Using the weighted aggregation reported above, the unineuron can be written as [85]:

$$\mathbf{z} = UNI(w; a; o) = U_{i=1}^{n}\, p(w_i, a_i) \tag{8}$$

where $T$ is a *t-norms* (product) and $s$ is a *s-norms* (probabilistic sum).

### 4.4. Third Layer

A singleton describes the third layer of the FNN model, that is, a single neuron capable of working as a universal function approximator [90]. Therefore, the answers obtained can predict future values linked to the climate. Consequently, this neuron present in the third layer can be seen from the following equation [13]:

$$y = \sum_{j=0}^{l} f_{linear}(z_l, v_l) \tag{9}$$

where $z_0 = 1$, $v_0$ is the bias, and $z_j$ and $v_j$, $j = 1, \ldots, l$ are the output of each fuzzy neuron of the second layer and their corresponding weight, respectively. $f_{linear}$ represents the linear activation function of the neuron, which results from the fuzzy input by the synaptic weight.

### 4.5. Training Algorithm

The model training algorithm is based on obtaining the neural network synaptic weights (and, consequently, the importance of the fuzzy rules of the inference system) through the concepts of using Moore Penrose's pseudo-inverse [91]. Thus, there is no requirement to update the parameters recursively, and, at the same time, there is a reduction in the influence of randomly defined parameters on the model structure. This technique allows weights to be set in one step. To solve possible overfitting problems in FNN training, a resampling regularization technique [92] is used in the objective function of training to define the neurons (or fuzzy rules) that contribute most efficiently to the model.

The values of the neural network weights of the hybrid model are qualified for the operationalization of the neuron assignments to implement the function approximation. When synaptic weights are defined analytically, the efficiency of the model is evidenced and favors obtaining fast and accurate solutions. In this paper, this vector is considered by the Moore–Penrose pseudo Inverse [13]:

$$\mathbf{v} = \mathbf{Z}^+ \mathbf{y} \tag{10}$$

where $y = [y_1, y_9, \cdots, y_n]^T$ is the desired output vector and $\mathbf{Z}^+$ is pseudo-inverse of Moore–Penrose [91] of $\mathbf{z}$. The $l+1$ dimensional input space ($\mathbf{z}$), generating a $n \times l + 1$ feature matrix, is presented as [13]:

$$\mathbf{z} = \begin{bmatrix} z_0 & z_1(x_1) & z_2(x_2) & \cdots & z_1(x_1) \\ z_0 & z_1(x_1) & z_2(x_2) & \cdots & z_1(x_1) \\ \vdots & \vdots & \vdots & \cdots & \vdots \\ z_0 & z_1(x_n) & z_2(x_n) & \cdots & z_1(x_n) \end{bmatrix} \tag{11}$$

In this view, $\mathbf{z}$ is the least norm of the least-squares resolution for the weights of the output layer. Basically, the purpose of learning is presented to find the $\mathbf{v}$ parameter that minimizes the error between the network output and wanted output for all training samples:

$$\sum_{i=1}^{n} \|z(x_i)\mathbf{v} - y_i\| \tag{12}$$

Since the number of second layer neurons may contain unnecessary information, it is essential to use analytical techniques to determine the relevance of the collected neurons to the problem analyzed. Statistical evaluations that attempt to discover the correlations between two vectors are performed by regularization techniques, such as Least-Angle Regression (LARS) proposed by Efron et al. [93].

LARS is a regression model for extensive data that can qualify and quantify exactly the regression coefficients as well as a part of candidate regressors to be associate in the final model. When refereeing a set of $n$ different samples $(x_i, y_i)$, the cost function of this regression algorithm can be presented to change Equation (12) [93]:

$$\sum_{i=1}^{n} \|z(x_i)\mathbf{v} - y_i\|_2 + \lambda \|\mathbf{v}\|_1 \tag{13}$$

where $\lambda$ is a regularization parameter, frequently evaluated by cross-validation.

The initial expression in Equation (13) corresponds to the Residual Sum of Squares (RSS). The relationship between the training error and the first term of Equation (13) is directly proportional so that the training error should be reduced by decreasing the RSS. The second is a $L_1$ regularization term. First, it develops FNN generalization, bypassing overfitting [94]. Secondary, it can be used to

produce sparse models [95]. To explain why LARS can be used as a neuron selection in the model, Equation (13) is rewritten as:

$$\min_{v} \quad RSS(v)$$
$$\text{s.t.} \quad ||v||_1 \leq \Gamma \tag{14}$$

where $\Gamma$ is an upper-bound on the $L_1$-norm of the weights. A small value of $\Gamma$ corresponds to a high value of $\lambda$, and vice versa. The model proposed by Bach [92], called Bolasso (Model Consistent Lasso Estimation through the Bootstrap), uses this methodology to find a combination of neurons that meet a previously defined criterion in a set of bootstrap replications.

This re-sampling approach is used to increase the stability of the model selection algorithm. Bolasso uses the LARS algorithm to operate on several bootstrap replications of the neurons in the second layer to perform model selection. For each repetition, a distinct subset of the regressors is selected. The neurons ($z_\Gamma$) to be included in the final model are defined according to the frequency with which each of them is chosen through different tests. A consensus threshold is determined, e.g., $\gamma = 80\%$, and thus a regressor is included, if selected in at least 80% of the assays.

Therefore, if a substantial number of bootstrap replications are chosen, there is a considerable probability that a fuzzy neuron will appear in the list of the most relevant ones. Thus, more relaxed selection criteria may determine a large number of candidate neurons at the end of the tests, just as a high $\gamma$ value may allow the bootstrap model to be rigid and select a small set of regressors. The fuzzy inference system acknowledgments can extract knowledge from a model-evaluated database, thus the rules obtained can be an element for developing expert systems. Thus, an example of a fuzzy rule set can be presented as follows [13]:

$$Rule_1 : \ If \ x_{i1} \ is \ A_1^1 \ with \ certainty \ w_{11}...$$
$$or \ x_{i2} \ is \ A_1^2 \ with \ certainty \ w_{21}...$$
$$Then \ y_1 \ is \ z_1$$

$$Rule_2 : \ If \ x_{i1} \ is \ A_2^1 \ with \ certainty \ w_{12}...$$
$$or \ x_{i2} \ is \ A_2^2 \ with \ certainty \ w_{22}... \tag{15}$$
$$Then \ y_2 \ is \ z_2$$

$$Rule_3 : \ If \ x_{i1} \ is \ A_3^1 \ with \ certainty \ w_{13}...$$
$$or \ x_{i2} \ is \ A_3^2 \ with \ certainty \ w_{23}...$$
$$Then \ y_3 \ is \ z_3$$

## 5. Meteorological Prediction Test

### 5.1. Evaluation Criteria

The evaluation criteria for time series follow peculiar characteristics that allow the validation of a model to be verified through an evaluation procedure that increases the prediction capacity of the model by the inclusion of new inputs. All collected data were normalized using zero means and the unit standard deviation to facilitate the validation by the models used in the tests.

The root means square error (RMSE) of the difference between the prediction and the real value is used as the evaluation criterion of the hybrid method. RMSE is a good measure because it usually explicitly represents what various methods tend to minimize. The RMSE is calculated as:

$$RMSE = \sqrt{\frac{\sum_{q=1}^{N}(y_q - \hat{y}_q)^2}{N}} \tag{16}$$

where $y_q$ is the response provided by the model (about the expected weather criterion) and $\hat{y}_q$ is the expected output for the test in question.

Another relevant criterion for predictor model analysis is the means square error (MSE), which is calculated by the following equation:

$$MSE = \frac{\sum_{q=1}^{N}(y_q - \hat{y}_q)^2}{N} \tag{17}$$

In addition to RMSE and MSE, the run-time factor was collected in the test. It is expressed in seconds. The tests were performed on a computer with the following settings: Core (TM) 2 Duo CPU, 2.27 GHz with 3-GB RAM.

### 5.2. Models Used in the Test

The models employed in the comparative experiment are the Linear Regression (LIN) [96], Gaussian Process (GAU) [97], and Multilayer Perceptron (MLP) [98]. The settings used in these models were: Linear Regression: attributeSelecion: M5Method, batchsize:100, ridge: $1.0 \times 10^{-8}$.
Multilayer Perceptron: batchsize = 100, hiddenlayer = a, learningRate = 0.3, momentum = 0.2, seed = 0, trainingTime = 500, validationSetSize = 0, validationThreshold = 20.
Gaussian Process: batchsize = 100, Kernel: polykernel, noise = 1.0, numDecimalPlaces = 2, seed = 1 are the originals maintained by the Weka software [99]. To perform the time series tests, the resources provided by the Weka Experimenter module were used, with 70% for training samples and 30% to obtain the model results. The fuzzy neural network used in this study obtained its parameters to be used in the experiments by choosing the best values in a 10 k-fold cross-validation procedure where the number of membership functions was chosen in the interval [2, 3, 4, 5], the number of bootstraps in [4, 8, 16, 32, 64], and the consensus threshold in [0.4, 0.5, 0.6, 0.7]. After preliminary procedures, the model was used in the final tests with two triangular membership functions, 16 bootstrap replications, and a consensus threshold of 0.5.

### 5.3. Rainfall Prediction Results

Forecasts may be compromised due to a dry period in the state between 2014 and 2016 [4]—a drought hit Brazil in that period causing water supply contingency in large Brazilian cities [100]. The drop in reservoirs harmed local businesses and agriculture [101] and allowed verifying the impact that the lack of water in the state can generate for social and economic aspects. Table 3 presents the comparative results in the rain prediction test for Belo Horizonte.

**Table 3.** Precipitation Prediction Test Results.

| Models | RMSE | Time | MSE |
|--------|------|------|-----|
| FNN | 60.45 | 47.63 | 3654.78 |
| MLP | 146.74 | 1.34 | 21,532.62 |
| LIN | 58.34 | 0.02 | 3403.53 |
| GAU | 64.51 | 0.05 | 4161.54 |

The results of the model training (Figure 17) show that the behavior of the monthly rainfall level in Belo Horizonte has very complex characteristics, an alternating moment of high peaks of water precipitation and months without rain. The model was able to identify the trends of this behavior, presenting very close to expected responses and mainly following the trends of increase and decrease of rainfall. The results obtained by the proposed model (FNN) (Figure 18) were satisfactory when compared to traditional approaches in the literature. Because it is an extremely complex problem, the model was able to identify crucial moments of behavior change, especially in the last period evaluated, where the state suffered from a massive water crisis.

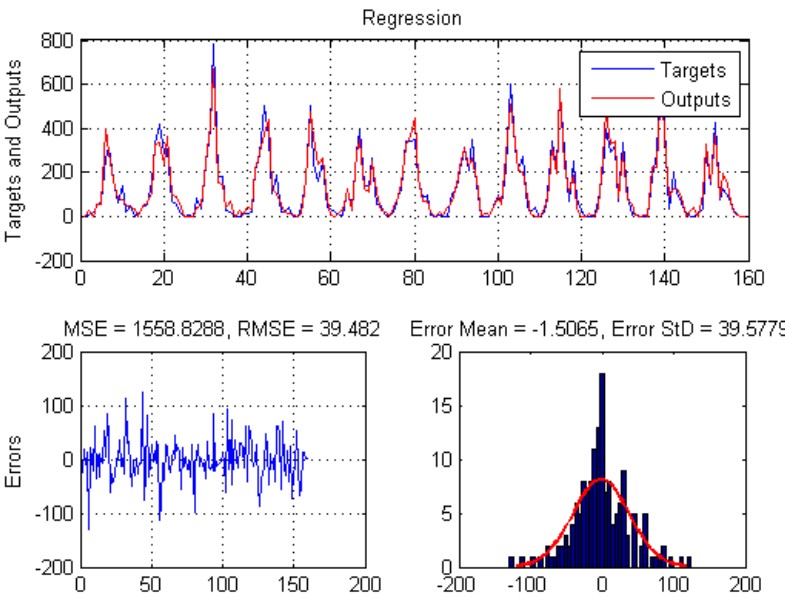

**Figure 17.** Hybrid model training phase: Rainfall.

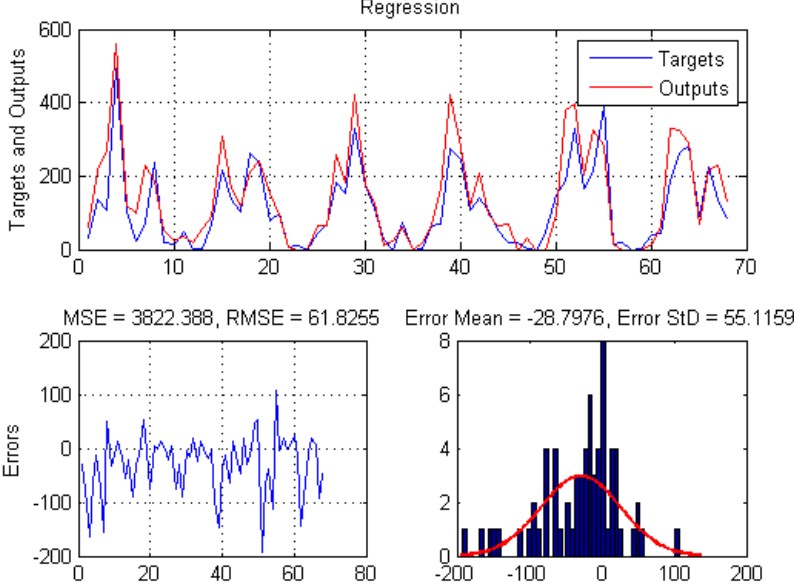

**Figure 18.** Hybrid model test phase: Rainfall.

### 5.4. Temperature Prediction Results

Another factor related to climate conditions in Belo Horizonte was evaluated by the hybrid model of fuzzy neural networks. Thus, the maximum temperature test results are presented in Table 4.

**Table 4.** Maximum Temperature Prediction Test Results.

| Models | RMSE | Time | MSE |
|--------|------|------|-----|
| **FNN** | 0.74 | 46.26 | 0.55 |
| **MLP** | 0.50 | 1.36 | 0.25 |
| **LIN** | 0.71 | 0.00 | 0.50 |
| **GAU** | 0.56 | 0.06 | 0.36 |

The fuzzy neural network model best identified the results obtained on the prediction of maximum temperatures in the state capital of Minas Gerais in the training phase (Figure 19). The model response data correctly identified the tendency for maximum temperatures to change during the analysis period. The final results of the model training presented answers with a low RMSE, but not so close to the traditional models (Figure 20). Its main advantage in this context is that the values are very close to reality, and the model can extract fuzzy rules from the characteristics used in this evaluation, as shown in Table 4.

In Figure 20, we can see that the model was able to understand the behavior of the data, performing an efficient approximation of the real value (comparison between blue line (expected) and the red line (model output). The closer the red line is to the blue line on the chart, the better the model predicted at the training and testing stages.

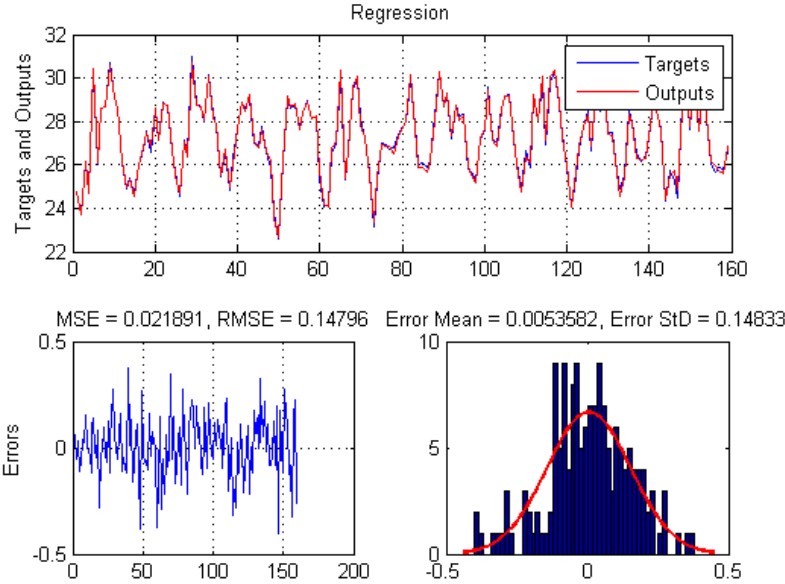

**Figure 19.** Hybrid model training phase: Temperature.

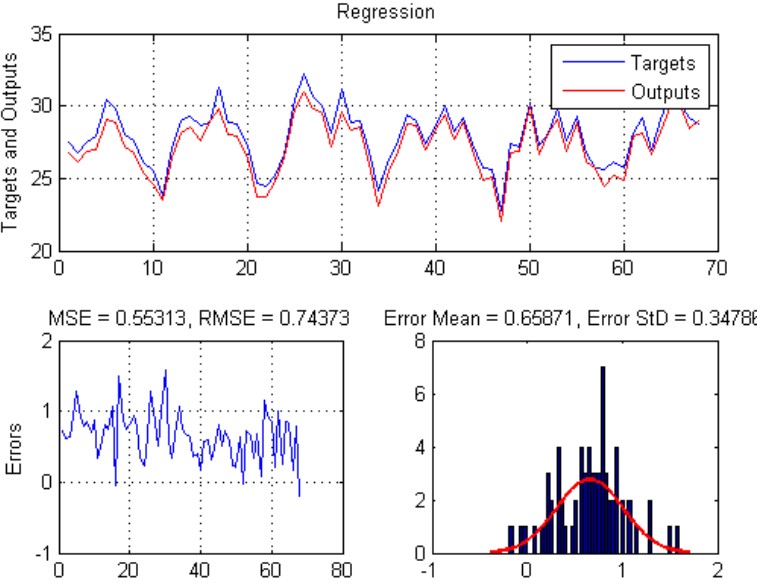

**Figure 20.** Hybrid model test phase: Temperature.

*5.5. Fuzzy Rules Generated*

The central feature of hybrid systems based on fuzzy systems and artificial neural networks is associated with the ability to extract fuzzy rules that can be viewed as a process of extracting knowledge from the data. These factors represent a correlated representation of the problems analyzed. As this paper aims to analyze the behavior present in meteorological data of the capital of the state of Minas Gerais, it is hoped that the rules found in these fuzzy relationships can support in processes of prevention of water shortage or actions related to awareness in the control of local temperatures. With fuzzy rules, knowledge can be more interpreted by people who are not experts in the field of artificial intelligence. IF/THEN statements can be seen as literal relations representing information. This factor becomes a differential to facilitate the acceptance and interpretation of discovered knowledge.

5.5.1. Temperature Prediction

The data collected by training and testing the model for the dimensions evaluated in the temperature prediction for Belo Horizonte city are listed, respectively, in Figures 19 and 20. As temperature variations are on smaller scales, the results obtained by intelligent models are more consistent with the responses obtained. Thus, the knowledge extracted from this database becomes more reliable. Moreover, one can analyze its aspects more realistically. Even during periods of uncontrolled environment, the maximum temperatures collected followed a specific behavior.

The following fuzzy rules were generated from the data obtained by the hybrid model. Thus, it is possible to identify more clearly the relationships between the features evaluated in the problem. These smart rules can make it easier to build intelligent systems, making it more accessible for people who do not comprehend artificial intelligence techniques to manage them.

1. If (Month is Final) and (Year is EndDecade) and (WindDirection is High) and (AverageWindSpeed is High) and (TotalInsolation is Low) and (AverageCloudiness is High) and (MeanPressure is Low) and (AverageCompensatedTemperature is Low) and (AverageMinimumTemperature is High) and (AveragerelativeHumidity is Low) and (TotalPrecipitation is High) then (TemperatureMaxAverage is 27.26).

2. If (Month is Final) and (Year is EndDecade) and (WindDirection is Low) and (AverageWindSpeed is High) and (TotalInsolation is High) and (AverageCloudiness is Low) and (MeanPressure is High) and (AverageCompensatedTemperature is High) and (AverageMinimumTemperature is Low) and (AveragerelativeHumidity is High) and (TotalPrecipitation is Low) then (TemperatureMaxAverage is 0.70).

3. If (Month is Final) and (Year is EndDecade) and (WindDirection is High) and (AverageWindSpeed is High) and (TotalInsolation is High) and (AverageCloudiness is High) and (MeanPressure is Low) and (AverageCompensatedTemperature is Low) and (AverageMinimumTemperature is High) and (AveragerelativeHumidity is Low) and (TotalPrecipitation is Low) then (TemperatureMaxAverage is $-0.11$).

4. If (Month is Final) and (Year is BeginningDecade) and (WindDirection is High) and (AverageWindSpeed is High) and (TotalInsolation is Low) and (AverageCloudiness is Low) and (MeanPressure is Low) and (AverageCompensated Temperature is High) and (AverageMinimumTemperature is Low) and (AveragerelativeHumidity is Low) and (TotalPrecipitation is High) then (TemperatureMaxAverage is $-0.07$).

5. If (Month is Initial) and (Year is EndDecade) and (WindDirection is High) and (AverageWindSpeed is High) and (TotalInsolation is High) and (AverageCloudiness is High) and (MeanPressure is High) and (AverageCompensatedTemperature is Low) and (AverageMinimumTemperature is Low) and (AveragerelativeHumidity is Low) and (TotalPrecipitation is High) then (TemperatureMaxAverage is $-0.11$).

6. If (Month is Final) and (Year is BeginningDecade) and (WindDirection is Low) and (AverageWind Speed is Low) and (TotalInsolation is High) and (AverageCloudiness is High) and (MeanPressure is Low) and (AverageCompensatedTemperature is High) and (AverageMinimumTemperature is High) and (AveragerelativeHumidity is Low) and (TotalPrecipitation is Low) then (TemperatureMaxAverage is $-0.13$).

7.   If (Month is Initial) and (Year is BeginningDecade) and (WindDirection is High) and (AverageWindSpeed is Low) and (TotalInsolation is Low) and (AverageCloudiness is Low) and (MeanPressure is High) and (AverageCompensatedTemperature is High) and (AverageMinimum Temperature is High) and (AveragerelativeHumidity is High) and (TotalPrecipitation is High) then (TemperatureMaxAverage is −0.29).

8.   If (Month is Initial) and (Year is BeginningDecade) and (WindDirection is High) and (AverageWindSpeed is Low) and (TotalInsolation is High) and (AverageCloudiness is Low) and (MeanPressure is Low) and (AverageCompensatedTemperature is Low) and (AverageMinimum Temperature is Low) and (AveragerelativeHumidity is Low) and (TotalPrecipitation is Low) then (TemperatureMaxAverage is 0.38).

9. If (Month is Final) and (Year is EndDecade) and (WindDirection is High) and (AverageWindSpeed is Low) and (TotalInsolation is Low) and (AverageCloudiness is Low) and (MeanPressure is Low) and (AverageCompensatedTemperature is High) and (AverageMinimumTemperature is High) and (AveragerelativeHumidity is Low) and (TotalPrecipitation is Low) then (TemperatureMaxAverage is 0.59).

10. If (Month is Initial) and (Year is EndDecade) and (WindDirection is Low) and (AverageWindSpeed is Low) and (TotalInsolation is Low) and (AverageCloudiness is High) and (MeanPressure is High) and (AverageCompensatedTemperature is Low) and (AverageMinimumTemperature is Low) and (AveragerelativeHumidity is High) and (TotalPrecipitation is Low) then (TemperatureMaxAverage is −0.26).

11.   If (Month is Initial) and (Year is BeginningDecade) and (WindDirection is Low) and (AverageWindSpeed is High) and (TotalInsolation is High) and (AverageCloudiness is Low) and (MeanPressure is Low) and (AverageCompensatedTemperature is Low) and (AverageMinimum Temperature is Low) and (AveragerelativeHumidity is Low) and (TotalPrecipitation is Low) then (TemperatureMaxAverage is 0.53).

### 5.5.2. Rainfall Prediction

The results obtained in the training and testing of the fuzzy neural network in predicting rainfall rates in the state capital of Minas Gerais are listed in Figures 17 and 18, respectively. As the rainfall level experienced extreme variations during the collected period, the model had some difficulties in identifying the actual values. However, it should be noted that, despite significant variations, the model was able to understand the trend of growth and decrease in rainfall rates. This feature is critical for models that work with time series. Thus, predictions can be estimated following the error raised by the experiments, where it should be highlighted that they are very close to other approaches traditionally used for this purpose.

The fuzzy neural network model highlighted the behavior of the various dimensions present in the problem (Figure 21). Rainfall rates increase as wind speeds increase, and we are in the early months of the year. Similarly, it is possible to analyze that, for the city of Belo Horizonte, there are high rainfall rates when a low insolation level high follows the cloudiness level in the city. Another critical factor is that rainfall in the city is concentrated to a greater extent when the maximum temperature in the city is around 24 °C, and the wind direction is high. At this maximum temperature and with high atmospheric pressure indices, there are also rain spikes in the state capital analyzed. Finally, in the 2000s, when there were high average wind speeds, rainfall rates were high for the city.

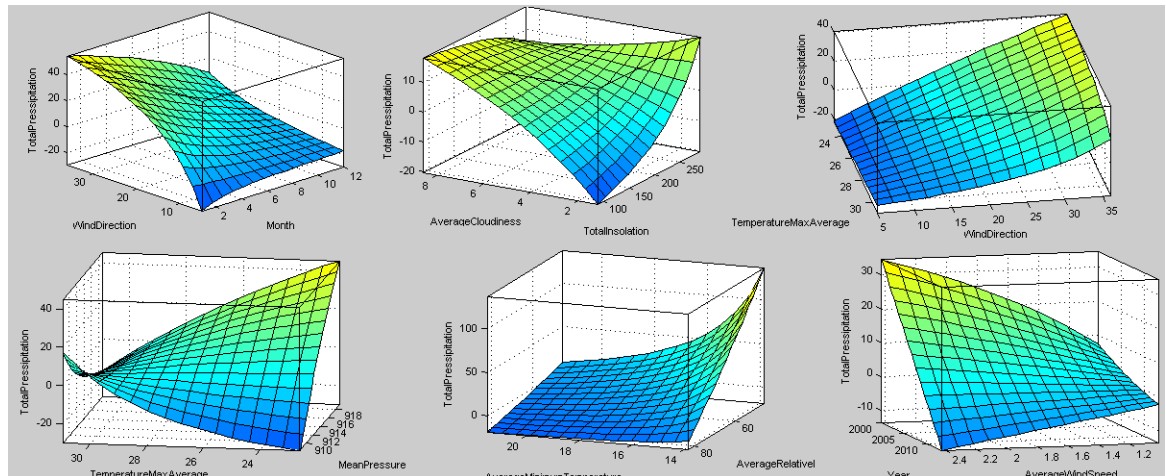

**Figure 21.** Decision space of the hybrid model. Rain predictions.

## 6. Conclusions

The use of fuzzy neural network models is feasible for detecting existing meteorological aspects in data collected through meteorological stations. This streamlines the joining of techniques from different sectors of science for the benefit of the population of the state of Minas Gerais. Thus, it is expected that, with the knowledge acquired in this scientific work, the control and monitoring of aspects related to rainfall and temperatures in the state become a factor in the planning of people and government of the state of Minas Gerais. As the economy and social welfare depend on a balance of the two factors analyzed, new technologies are expected to be incorporated to improve control and tracking of weather stations, thus enabling citizens to receive predictive information to prevent or control certain behaviors that harm the environment.

The rules obtained by the intelligent system are entirely based on the nature of the data present in the routine of state residents. As it was visualized that the climatic characteristics are changing over time, it is expected that, in the future, these impacts will be more substantial, mainly with an increase of average temperatures and a decrease of the rainfall indices. Managers and the public need to prepare for such impacts while seeking ways to mitigate the impacts on industry and people's daily lives. After validating the fuzzy rules obtained in the paper by professionals in the field, they can be used for the construction of expert systems to predict weather factors.

Future work can investigate weather forecasting factor acknowledgments with missing data, using algorithms capable of controlling them in the assessment. Other hybrid models, neural networks models, feature selection techniques, and resampling missing data can be leveraged for pertinent extensions of this paper.

**Author Contributions:** Conceptualization, P.V.d.C.S. and L.B.d.O.; methodology, L.A.F.d.N.J.; software, P.V.d.C.S. and L.A.F.d.N.J.; validation, L.B.d.O. and L.A.F.d.N.J.; formal analysis, P.V.d.C.S.; investigation, L.B.d.O.; resources, P.V.d.C.S.; data curation, P.V.d.C.S. and L.A.F.d.N.J.; writing—original draft preparation, P.V.d.C.S. and L.B.d.O.; writing—review and editing, P.V.d.C.S. and L.B.d.O.; visualization, L.A.F.d.N.J.; supervision, P.V.d.C.S.; project administration, P.V.d.C.S.; and funding acquisition, P.V.d.C.S. and L.A.F.d.N.J.

**Funding:** This research received no external funding.

**Acknowledgments:** The authors acknowledge the Faculty Una of Betim and the Federal Center of Technological Education of Minas Gerais—CEFET-MG.

**Conflicts of Interest:** The authors declare no conflict of interest.

## Abbreviations

The following abbreviations are used in this manuscript:

MDPI   Multidisciplinary Digital Publishing Institute
MG     Minas Gerais
FNN    Fuzzy Neural Network
UNI    lUnineuron

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
