# Peer review of "Fuzzy Rules to Help Predict Rains and Temperatures in a Brazilian Capital State Based on Data Collected from Satellites"

_applsci, doi:10.3390/app9245476_

Round 1

Reviewer 1 Report

Line 11 yours, replace with their

Line 31: we have problems : the state of belo horizonte , or whatever location, has problems, not the authors.

Line 36: what is rules nebulae . Also a period is missing before Developing

Line 41 Grammar . Are being developed

Line 42: Due, capital D after period , or should be a comma ? Please do a serious revision of English before submitting a paper for reviewing.

Paragraph starting in line 41 and rest of the manuscript: In English, technical papers should be written in third person singular (it), not in first person plural (we) as it shows in the paper, unless you use WE to refer to the actions or findings from the authors.

Line 41: artificial neural networks (ANN), and then you  can use ANN in the following sentences.

The state of the art starting in line 41 should be limited to rainfall and temperature, the topics of your paper. The way it is written seems to try to address “rain forecast, river basins, water studies, and so on” but very vaguely.

Lines 61 to 64 , need some rewording

Line 78: In English are called membership functions, not pertinence

Line 106: the mining territory: replace by the territory. The figure don’t show “mining” operations.

Lines 126 to 132. Unnecessary. Remove

I don’t see how section 1.2 and Figure 3 add relevant content to the manuscript.

Figure 2: I would suggest having a figure showing the river network of Minas Gerais, which is what you are describing in section 1.1

The numbering of the sections is incorrect. Section 1 should be the introduction after the abstract.

Author Response

Dear Reviewer.
Thanks for the valuable tips. Thank you so much for being so thorough. I believe the paper gained a lot in quality with its review.
Then the answers to the questions, informing that the changes are highlighted in the paper in red.

Line 11 yours, replace with their

Answer: Thanks for the tip. It was properly changed.

Line 31: we have problems : the state of belo horizonte , or whatever location, has problems, not the authors.

Answer: Thanks for the tip. It was properly changed.

Line 36: what is rules nebulae . Also a period is missing before Developing

Answer: Thanks for the tip. It was properly changed. The sentence has been rewritten

Line 41 Grammar . Are being developed

Answer: Thanks for the tip. It was properly changed. 

Line 42: Due, capital D after period , or should be a comma ? Please do a serious revision of English before submitting a paper for reviewing.

Answer: Thanks for the tip. It was properly changed. The sentence has been rewritten

Paragraph starting in line 41 and rest of the manuscript: In English, technical papers should be written in third person singular (it), not in first person plural (we) as it shows in the paper, unless you use WE to refer to the actions or findings from the authors.

Answer: Thanks for the tip. The terms were duly replaced throughout the entire text.

Line 41: artificial neural networks (ANN), and then you can use ANN in the following sentences.

The words really were very repetitive and the text very tiring. The paragraph has been rewritten.

The state of the art starting in line 41 should be limited to rainfall and temperature, the topics of your paper. The way it is written seems to try to address “rain forecast, river basins, water studies, and so on” but very vaguely.

Answer: Dear Reviewer. Thanks for the tip. The text has been changed to make rain forecasts more prominent in various contexts.

Lines 61 to 64 , need some rewording

Answer: Dear Reviewer. Thanks for the tip. The paragraph has been rewritten to make reading simpler.

Line 78: In English are called membership functions, not pertinence

Answer: Thanks for the tip. It was properly changed. 

Line 106: the mining territory: replace by the territory. The figure don’t show “mining” operations.

Answer: Dear Reviewer. Thanks for the tip. It was a translation error. The sentence has been duly corrected.

Lines 126 to 132. Unnecessary. Remove

Answer: Dear Reviewer. Thanks for the tip. The text has been properly removed.

I don’t see how section 1.2 and Figure 3 add relevant content to the manuscript.

Answer: Dear Reviewer. Thanks for the comment. I think it is relevant to highlight the change in the direction of the winds, with emphasis on Brazil and consequently on the state of Minas Gerais. One of the collected dimensions is linked to wind speed.

Figure 2: I would suggest having a figure showing the river network of Minas Gerais, which is what you are describing in section 1.1

Answer: Dear Reviewer. Thanks for the tip. A highlight the watersheds and hydroelectric plants in the territory of the state of Minas Gerais were properly placed in figure 1.

The numbering of the sections is incorrect. Section 1 should be the introduction after the abstract.

Answer: Dear Reviewer. Excuse me. It was an error in the final formatting of the paper. The paragraphs have been duly corrected.

Reviewer 2 Report

Dear Authors

The effort made by authors is very useful and meaningful in the scientific community implied by the title and abstract. However, the paper contains several weak parts of the research. Focusing on the broader audiences, the required amendments are described in the enclosed in the attached file, I strongly suggest the authors have a major revision. I look forward to seeing the effort of the authors in the future.

Sincerely

Anonymous Reviewer.

Author Response

General comments

The presented research by Souza et al. is very interesting and practical research especially when the planet is undergoing climate change. The broader audience will attract profoundly just by looking at the title. The tricks presented in this manuscript seem to be very useful and can be mimicked anywhere upon the availability of data.

Dear Reviewer. Thank you for the compliments.

However, there exists a substantial amount of improvement that can be done prior to the publication. While the title emphasizes the prediction of rainfall and temperature, the temporal scale used in this study is not clear now. Even though the authors mentioned the concept of the hybrid model in the beginning, the results from the model are not clear. The evaluation of the model in the training phase is not well presented.

Dear Reviewer. Thanks for the comments. We modified the structure of the text according to the advice of the reviewers.

There is a sufficient amount of literature cited throughout the paper, but the objective and gaps that filled by this study are not clear. This manuscript needs English writing services. There are many places where sentences are too long and with a lot of punctuation errors. Especially the introduction section needs to be well rewritten.  Because of errors in semantic, the introduction section is hard to follow while reading.

Dear Reviewer. Thanks for the comments. We fully agree that the introduction needed to be redone. In the revised version, one can notice the amount of changes that were made in the introduction. We believe the text has become more understandable.

Sections 3.2 to 3.5 are mathematically rich ones. Nevertheless, the symbols are not well defined. For the general audience with limited knowledge of machine learning, the mathematical formula is hard to understand.

Dear Reviewer. Thanks for the comments. The symbols were better specified to make the reading richer and more understandable.

The presentation of results is very poor. The Figures from 17-20 do not have axes labeling and other components are not greatly presented as well. Figure 21 seems to be redundant. The related explanation related to this Figure is not relevant for policymakers. There is a miss location of Tables 3 and 4, which are prior to the text. Moreover, the caption of the figures and tables is too short. For example, the caption of Figures 14 and 15 should be elaborate more. It seems there is redundant information. The embedded text is not clear enough to visualize.

Dear Reviewer. Thanks for the comments. The results were better explained. Explanations of the figures were also performed.

The conclusions made in the manuscript seem to be over-optimistic. The reviewer does not see potential applications at the operational level rather useful to visualize complex processes. For instance, the statement made in line 599 "... the nature of the data ..." may not be realistic

Answer: Dear Reviewer. Thanks for the comment. We believe that knowledge extraction and process automation can result in better controls by managers in the state of Minas Gerais. In developing countries, AI is not yet so widespread. Therefore, this model can act assertively, extracting knowledge from a simpler way to be understood by administrative managers, who often do not have access to AI terms. These results, being the first, can be precursors of new systems, new analyzes and new experiments.

Given the poor quality of presentation despite appreciating the great effort of authors, the reviewer afraid to accept the manuscript to publish in the present form. In order to make a decision, the authors should greatly emphasize improving the usage of English and then the presentation of results in a better way so that text and contents of the tables and figures are matched as well.

Dear Reviewer. Thanks for the comments. His assessment and points were important for improving the paper. Following are the comments made on each of the suggestions raised by you.

Highlights of the changes are highlighted in red in the manuscript.

In lines 21-22, it will be better to say "predictions of natural disasters like floods, droughts" instead of "predictions of floods, droughts, and natural disasters" Dear Reviewer. Thanks for the sugestion. It has been duly amended in the text. Revise sentence in lines 36-39. At this moment, the meaning of the sentence is not clear. Dear Reviewer. Thank you. The paragraph has been rewritten to have smaller and more meaningful sentences. In line 40, the authors mentioned, "it is also possible to anticipate solutions, and reduce their impacts." However, it is not clear what the solution is and how impacts are reduced. Dear Reviewer. Thanks for the tip. An example has been added to the text. In line 87, there should be section 2 instead of section (1) and this should be entitled as "Introduction." Dear reviewer. Problem fixed. What are the objective models mentioned in line 145? Dear Reviewer. It was a typo that has already been properly corrected. The sentence in lines 182-184 "To represent the behavior of climatic aspects ... and ... between 1931 and 1960 ... 1961 to 1990 shall be revised. Right now, the sentence is not correct. Dear Reviewer. The term month by month has been added to specify the correct period. What do "feet" and "tip" of the curve mean? For non-expert audiences, this jargon appeared in Equation 4 needs to be explained. Dear Reviewer. Thanks for the tip. The terms were better rewritten. What is the proper form of . Define it properly. Dear Reviewer. Thanks for the tip. The terms were better rewritten. The results presented in Figures 17-20 are not clear to understand whether the result belongs to which model out of 4 models listed in Tables 3 and 4. The four models used in the test are to verify that the FNN results are better than state of the art. Thus, it is only interesting to highlight the output of the fuzzy neural network. The fuzzy rules generated in Section 4.5.1 seem to be a confusing lot. one way to make it more clearly is to have some tree. Another issue raised for the middle month. Introducing a decade makes the reader confuse the temporal resolutions of analysis as well.

Dear Reviewer. Thanks for the tip. However, this type of model creates rules of type TKS. These rules are the presented result of the model, based on equation 15.

Specific comments

In line 6, an article "the" is missing before the word, "third". Dear Reviewer. Thank you. The term has been properly inserted/fixed. In line 9, "as" is not a good word; it should be replaced by an article like "a". Dear Reviewer. Thank you. The term has been properly replaced. In line 11, what the authors want to say with the word "your". Dear Reviewer. Thank you. The term has been properly inserted/fixed to their. In line 16, "indexes" should be replaced by "indices". Dear Reviewer. Thank you. The term has been properly replaced. In line 20, "where" is not an appropriate word. Such could be replaced by "because" or "as". Dear Reviewer. Thank you. The term has been properly replaced by as. The sentence in lines 19-23 is too long. Dear Reviewer. Thank you. The paragraph has been rewritten to have smaller and more meaningful sentences. The sentence in lines 24-26 is not clear and there are many semantic errors in regards to clauses. Dear Reviewer. Thank you. The paragraph has been rewritten to have smaller and more meaningful sentences. In line 26, "comma" is missing after the word "years". Dear Reviewer. Thank you. The term has been properly inserted. In line 36, "period" is missing after "nebulae". Dear Reviewer. Thank you. The paragraph has been rewritten to have smaller and more meaningful sentences. In line 39, "comma" is missing after "Briefly". Dear Reviewer. Thank you. The term has been properly inserted. In line 42, "due" should be "Due" - punctuation error. Dear Reviewer. Thank you. The paragraph has been rewritten to have smaller and more meaningful sentences. Revise the phrase "as an example" in line 43. Dear Reviewer. Thank you. The paragraph has been rewritten to have smaller and more meaningful sentences. In line 45, ";" should be "." to reflect the start of another sentence. Dear Reviewer. Thank you. The paragraph has been rewritten to have smaller and more meaningful sentences. A "period" needs to replace a "semicolon" inline 47. Dear Reviewer. Thank you. The paragraph has been rewritten to have smaller and more meaningful sentences. A "period" needs to replace a "semicolon" in line 51. Dear Reviewer. Thank you. The paragraph has been rewritten to have smaller and more meaningful sentences. A "period" needs to replace a "semicolon" in line 53. Dear Reviewer. Thank you. The paragraph has been rewritten to have smaller and more meaningful sentences. In line 55, the word “this" seems to be redundant. Dear Reviewer. Thank you. The paragraph has been rewritten to have smaller and more meaningful sentences. A "period" needs to replace a "semicolon" in line 58. Dear Reviewer. Thank you. The paragraph has been rewritten to have smaller and more meaningful sentences. In line 104, "1.500" should be replaced by "1500". Dear Reviewer. Thank you. The term has been properly fixed. The word "waters" in line 115 should be corrected as "water." Dear Reviewer. Thank you. The term has been properly fixed. In line 121, "indexes" should be replaced by "indices". Dear Reviewer. Thank you. The term has been properly fixed. In line 141, revise the words "consequently" or "consequences" appropriately. Dear Reviewer. Thank you. The term has been properly fixed. In line 200, "indexes" should be replaced by "indices". Dear Reviewer. Thank you. The term has been properly fixed. It is not clear that numbers within the parenthesis in Table 2. Dear Reviewer. Thanks for the tip. This is the standard deviation. The following text has been added: Where applicable, the standard deviation is given in parentheses. The symbols like,, and others are not properly defined in Equations 1 and 2. Dear Reviewer. Thank you. The terms have been properly explained. In line 366, "or" is redundant at the end of the sentence. Dear Reviewer. Thanks for the tip, but that's correct. There is the and neuron and the or neuron. In line 486, the first letter of Table 4 should be capital. Dear Reviewer. Thank you. The term has been properly fixed. In line 507, the first letter of Figures shall be capital. Dear Reviewer. Thank you. The term has been properly fixed.

Round 2

Reviewer 2 Report

After reviewing the revised version, the reviewer, first, thanks to the authors’ response in a positive way. It is clear that the authors genuinely fixed the technical issues. Even though, some grammatical errors need to be taken care of before the publication.

To the end, the reviewer congratulates the authors and hope to see much improvement in the final version.